# Estimation of energy efficiency of heat pumps in residential buildings using real operation data

Tobias Brudermueller [1] ✉, Ugne Potthoff [1], Elgar Fleisch [1,2], Felix Wortmann [2] ✉ & Thorsten Staake [1,3]

As heat pumps become more prevalent in residential buildings, effective performance monitoring is essential. Design flaws, incorrect settings, and faults can escalate energy consumption and costs, leading to discrepancies in user expectations and hindering the widespread adoption of this technology crucial for the heating transition. However, field studies using large data sets to offer insights into real-world performance and methods for identifying low-performing systems in practical, scalable applications are lacking. In the largest field study to date, we analyze sensor data from 1023 heat pumps across Central Europe monitored over two years. Based on existing approaches for controlled laboratory conditions, we derive methods to evaluate and classify real-world performance using operational data. Applying these methods, we find that 17% of air-source and 2% of ground-source heat pumps do not meet existing efficiency standards. Additionally, around 10% of systems are oversized, while approximately 1% are undersized. This underscores the need for standardized post-installation performance evaluation procedures and digital tools to provide actionable feedback for users and installers to enhance operational efficiency and guide future installations.

Buildings constitute 30% of global final energy consumption and contribute to 26% of global energy-related carbon dioxide emissions, with approximately half attributed to space and water heating[1]. Electrically-powered heat pumps (HPs), extracting heat energy from natural sources such as the ground, air, or water, offer a sustainable alternative to oil or gas-based heating, especially in regions with a high share of renewable electricity generation[2]. While already meeting 10% of global space heating needs in 2021, HPs have the potential to reduce global carbon dioxide emissions by at least 500 million tonnes by 2030, equivalent to the annual emissions from all cars currently in operation in Europe[3]. Yet, meeting the International Energy Agency's global non-binding target of 600 million HPs by 2030 necessitates an accelerated deployment of HPs, as current installation rates project a 58% shortfall[4]. The replacement of fossil fuel heating systems however poses a significant financial challenge for homeowners due to the high upfront costs, even with available subsidies and the potential for better cost amortization through achieved savings[5,6]. Subsidies are prevalent in over 30 countries[1] but also represent a substantial financial burden for governments[4,7,8]. Moreover, regulations regarding heating systems in private households are sometimes entangled with emotional responses[9]. The German Building Energy Act serves as a notable example, reflecting public discontent arising from being compelled to invest in heating renovation[5,10–12].

Additionally, HPs exhibit greater complexity compared to well-established gas and oil heating systems, and unlike these traditional systems, they have not undergone decades of optimization. The performance of HPs is influenced significantly by factors beyond design, such as occupant characteristics and HP system settings[13–17], which is a

[1]Chair of Information Management, ETH Zurich, Weinbergstrasse 56/58, Zurich, Switzerland. [2]Institute of Technology Management, University of St. Gallen, Dufourstrasse 40a, St. Gallen, Switzerland. [3]Chair of Information Systems and Energy Efficient Systems, University of Bamberg, An der Weberei 5, Bamberg, Germany. ✉e-mail: tbrudermuell@ethz.ch; felix.wortmann@unisg.ch

challenge for manufacturers, installers and owners. Consequently, the actual energy consumption of HPs in practice can deviate significantly from expectations, resulting in substantial additional operating costs. For example, Nolting et al.[15] report up to 24% lower performance than stated on the product certificate label. Additionally, in an analysis of 297 Swiss households with HPs, Weigert et al.[18] demonstrated that after on-site optimization by an energy consultant, half of them achieved average savings of 1805 kWh (15.2%) per year. As the operating costs of HPs are decisive in determining whether the technology is economically attractive compared to other heating solutions[19–21], the discrepancy in performance can fuel dissatisfaction and, ultimately, pose a threat to the technology's acceptance[22]. The success of the heating transition is therefore closely tied to the performance of heat pumps in the field.

Maximizing the energy efficiency of HPs is also relevant for electricity grids, as these systems significantly increase both total and peak power demand. This heightened demand can necessitate the implementation of demand response programs and expensive upgrades to grid infrastructure[23,24]. For example, transitioning 10% of British households to HPs would increase peak demand by 2.5–3.75 GW (4.6–7.0%)[25,26], and a transition of all British households would double it[27]. Similar studies are also available for other countries and scenarios[28–32]. In Switzerland, even with substantial already existing pump storage facilities that help control demand, a 34% increase in electricity storage capacity would be necessary if all fossil-based heating systems were replaced by HPs[29].

Digitalization offers opportunities to tackle the current challenges of HP operations. As the majority of modern HP units are equipped with multiple sensors providing real-time data, it becomes possible to monitor their performance and control their operation effectively[24,33,34]. However, manufacturers are still in the early stages of offering services beyond displaying raw consumption data and activating alarms in case of heating failures[35]. While fault detection and diagnosis systems have been thoroughly explored in the literature[36], there is a notable gap in research focusing on HPs operating without faults but potentially lacking optimization. Moreover, many existing studies lack the analysis of large sample size data sets of HPs in real-world settings, often relying on data from test houses, simulations, or laboratory experiments. For instance, Carroll et al.[37] conducted a structured literature review of field studies on air-source heat pumps and identified only 34 articles, with[38] being the largest field study with a sample size of 77. A more expansive and frequently cited field study was conducted by the UCL Energy Institute between 2013 and 2015, encompassing 292 air-source heat pumps and 92 ground-source heat pumps in the UK as part of the Renewable Heat Premium Payment (RHPP) program[39]. In addition to the limited number of field studies, there is a scarcity of studies developing methods to identify low-performing HPs in practical applications. Offering users individualized feedback about the energy efficiency of their HPs, however, has proven to significantly impact user satisfaction and can substantially enhance the acceptance of the technology[22]. Therefore, to improve the energy efficiency and reduce operational costs of HPs in real-world scenarios, gaining a profound understanding of their current performance and identifying systems with optimization potential, is imperative.

Despite this research gap, relevant studies in related domains can be classified into three categories: comparing real-world performance to product certificates, identifying underperforming states in individual systems, and developing optimal control strategies for individual systems. The first group of work compares in-situ performance of small sample sizes with corresponding benchmarks given in data sheets and finds significant differences between laboratory and on-site performance, e.g.,[15,40,41]. As explained by O'Hegarty et al.[41], European regulations define fixed operational points for the evaluation and

reporting of performance in standardized product certificates. Therefore, the contribution of this type of related work lies mainly in the development of complex interpolation and extrapolation methods to compare real-world performance to product certificates. Although this type of study is able to identify HPs where performance largely deviates from specifications, it requires contextual information (e.g., the exact model of the HP) and highly detailed measurements, which makes it unsuitable for mass market applications. In contrast, the second category examines performance of individual systems without further reference, e.g.,[42–45]. This group of studies describes performance under perfect knowledge of the buildings and without comparison of HPs in large populations. If methods for detecting underperformance are proposed (e.g., in[44]), the focus is on detecting periods of inefficient operation or performance degradation of single systems. Lastly, the third group of work targets optimal control strategies for individual systems to improve their energy performance, e.g.,[46–49].

The studies outlined above are not applicable to practical scenarios, as they fail to address variations in data availability, observation periods, and building characteristics. Moreover, these studies do not offer methods for identifying low-performing HPs within their specific installation environments. In practical settings, relevant contextual information such as building type, occupancy levels, and more, is typically unavailable. Additionally, HPs may not be controllable to operate precisely at the operational points defined in regulations, thereby hindering comparisons to known performance values derived from idealized laboratory conditions. This lack of control may be due to technical constraints or concerns about regular adjustments impacting occupant comfort. As a result, evaluating HP performance in real-world applications must rely solely on observations of actual operational conditions, without access to detailed contextual knowledge or the ability to directly interfere with its operation. Such assessments are critical in practice and are highly desired by HP users[22], yet effective methods for conducting them have not been established previously. Additionally, clear benchmarks for good and poor performance in practical applications are lacking.

In this study, we address the existing gap by developing methods to evaluate HP performance in the field post-installation using real operational data. Additionally, we provide insights from a comprehensive performance analysis of 1023 HPs installed in residential buildings. Our findings reveal significant variability in performance among individual HPs, with a 2-3 fold difference between the lowest and highest efficiency systems. Moreover, 17% of air-source and 2% of ground-source HPs fall short of existing European efficiency standards. Approximately 10% of systems are oversized, while about 1% are undersized. These results highlight the critical need for standardized post-installation performance evaluation procedures and the development of digital tools to deliver actionable feedback for users and installers, ultimately improving operational efficiency and informing future installations.

## Results
### Real-world data set
The analyzed data encompasses a wide variety of HP models and configurations installed in residential buildings across 10 countries in Central Europe. Since the data originates from a single manufacturer, we acknowledge that our results should be further validated with data from HPs produced by other companies to ensure broader applicability and generalizability. Nonetheless, this study represents the largest field study conducted to date on the energy efficiency of HPs in residential buildings. The data set studied covers 1,023 HPs monitored between 2021-03-14 and 2023-04-30 (i.e., for up to 777 days), with 890 (87%) being air-to-water HPs and 133 (13%) brine-to-water HPs. There are no water-to-water HPs in the data set. While the descriptive

analyses encompass all systems, other analyses are limited to HPs with appropriate data to avoid distortions from poor model fits. Most analyses include around 600 to 700 systems, with the exact number of samples reported in each subsection. While other contextual information is unavailable, it is known that all HPs are installed in residential buildings in Germany: 434 (42.42%), the Netherlands: 211 (20.63%), Austria: 204 (19.94%), Czech Republic: 78 (7.62%), Sweden: 46 (4.50%), Denmark: 35 (3.42%), Poland: 3 (0.29%), Slovenia: 2 (0.20%), France: 1 (0.10%), Great-Britain: 1 (0.10%), and unknown: 8 (0.78%). In total, the data set contains 185,139 daily observations at outdoor temperatures of 15 °C or below, with each HP having an average of 182.22 days of data within this temperature range.

Each HP is connected to the internet and measures multiple parameters with a temporal resolution of a few seconds. To reduce complexity, in this study, we analyze daily aggregates of this data, using daily sums for electrical energy consumption and thermal energy production, and daily averages for all other parameters. Days with data gaps have been systematically excluded to ensure the highest attainable data quality and are not counted in the number of observations mentioned above. Days with more than three hours (12.5%) of missing measurements in total are removed. Further, any day lacking measurements of outdoor and supply temperatures, energy input, or energy output is neglected. Furthermore, measurements are categorized based on the operating modes for domestic hot water (DHW) production and space heating (SH). Since all HPs are used for SH but not all are used for DHW production, our analyses focus on SH to ensure comparability. The performance metrics reported in this study encompass final energy usage of the compressor, the fan or brine pump, and the electrical backup heater. This aligns with the European standard EN 14825[50] and adheres to the $H_3$ system boundary taxonomy outlined in[41,51]. Further, it is worth noting that the energy values are not directly measured by energy sensors; rather, they are computed by the HPs themselves using operational sensors and the principles of physics. This computation relies on parameters such as pressure, volume flow, and power measurements and is common practice in most modern HPs[15].

Due to potential errors in measurements, particularly during minor compressor modulations, inaccuracies in the recorded HP performance can occur. The specifics of the sensors used are unavailable to us, precluding detailed calculations of measurement uncertainties. However, a draft proposing updates to regulations by the European Union concerning HPs[52] suggests that the maximum permissible error for energy output should range from 7.5% to 15%, depending on the temperature difference. For energy input, a maximum permissible error of 5% is proposed. According to the HP manufacturer, the errors in the data under study already fall within these tolerances.

## Modeling and evaluating heat pump performance

This section provides essential fundamentals to support the descriptions of the results and outlines the methods developed in this study, which are designed for post-installation performance evaluation in practical applications.

**Explaining Carnot efficiency.** Our analyses rely on the coefficient of performance (COP) as a key metric for evaluating the efficiency of an HP, which is the ratio of thermal energy generated to electrical energy consumed in a fixed observation period. This metric is not constant and affected by operational conditions, as comprehensibly reviewed in[53]. The maximum efficiency theoretically achievable by an HP is defined by the Carnot cycle and depends on the difference between the heat source temperature $T_{hsource}$ and the heat supply temperature $T_{hsupply}$ in Kelvin. In practice, however, HPs typically operate at around half of their theoretical maximum efficiency or even lower, influenced by irreversible and non-ideal effects extensively studied in the

literature[42,54,55]. These effects can be represented by a correction factor $\zeta$, defining the COP as:

$$COP = \zeta \cdot \frac{T_{hsupply}}{T_{hsupply} - T_{hsource}} \tag{1}$$

From Equation (1), it can be inferred that HPs are efficient when the temperature difference is small, achieved by using low flow temperatures for the water distributed by the HP to a space or system[40]. An in-depth analysis of the underlying reasons for the values of $\zeta$ in practical applications is not the focus of this study. However, a comparison of observed COP values with Carnot efficiency and other models found in the literature is provided in Supplementary Note 2. Additionally, we note that several other factors are known to affect HP performance, such as the frequency of on-off transients[14], the quantity of defrosting cycles[53], the speed of the compressor[56], and variations in temperature profiles[44] or part-load conditions[57].

**Explaining part-load ratio and capacity ratio.** The performance of HPs, as reported in product certificates, assumes a fixed part-load ratio (PLR) at different operating points. The PLR is the ratio of the heating load at a specific temperature ($T_j$) to the design heating load at the design temperature ($T_{design}$), under the assumption of a linear relationship with outdoor temperature above the heating limit temperature ($T_{lim}$)[58]. According to EN 14825[50], for an average climate, $T_{design}$ is assumed to be -10 °C, for warmer climate it is 2 °C and for colder climate it is -22 °C. The heating limit temperature $T_{lim}$ is assumed to be 16 °C. As formulated by Sieres et al.[58], the PLR is given by:

$$PLR(T_j) = \begin{cases} (T_j - T_{lim})/(T_{design} - T_{lim}) & \text{if } T_j < T_{lim} \\ 0 & \text{if } T_j \geq T_{lim} \end{cases} \tag{2}$$

Another metric, closely related but not identical to the PLR, is the HP's capacity ratio (CR). While the PLR is independent of the HP capacity, the CR represents the HP's output capacity at $T_j$ relative to its full load capacity. Consequently, depending on the design choice of the HP's full load capacity, the CR line may lie above or below the PLR line, but likely remains close to it.

**General approach for modeling HP performance.** Data availability can vary significantly among HPs in terms of observation periods and operation at different temperatures. Therefore, we assess and ensure comparability of HP performance by modeling each system's behavior and performance based on its in-situ measurements, facilitating simulation and evaluation. We accomplish this by fitting linear mixed-effects models, which include fixed effects for all HPs (including slope and intercept) and individual random effects (also including intercept and slope). These random effects capture the individual deviations of each HP from the mean of all systems. In the following sections, we denote a parameter associated with random effects using a superscript $i$, where $i$ indexes a specific HP. As we proceed with the models defined below, several models based on existing literature were tested, with detailed results presented in the Methods section. Further note that the models are fitted and evaluated using observations from the SH mode exclusively, which is the primary application of focus (i.e., DHW is not included). The fitted model parameters for each individual HP are provided as supplementary material, enabling future studies to conduct simulations based on real-world data rather than product certificates. Finally, it is important to clarify that all subsequent models are fitted using the complete data available for each HP. However, to ensure robustness, additional tests were conducted by splitting the data for each system into training and test sets (see Supplementary Note 1). The model performance was then evaluated solely on the test data that was not seen during training. Since the differences in model fits however were minimal, we chose to use the models fitted with the

**Table 1 | Mean and standard deviation scores (in brackets) of the fits for individual models**

| Model Type | MdAE | MAE | MSE | RMSE | MAPE | SMAPE | $R^2$ |
|---|---|---|---|---|---|---|---|
| Heating Curve | 1.02 (0.69) | 1.20 (0.75) | 2.99 (4.58) | 1.48 (0.90) | 3.32 (2.06) | 3.30 (2.02) | 0.43 (0.27) |
| COP | 0.19 (0.11) | 0.24 (0.13) | 0.12 (0.14) | 0.30 (0.18) | 5.65 (2.48) | 5.58 (2.38) | 0.75 (0.17) |
| Heating Curve & COP | 0.23 (0.13) | 0.28 (0.15) | 0.16 (0.18) | 0.36 (0.19) | 6.63 (2.79) | 6.66 (2.83) | 0.51 (0.31) |
| Utilization | 4.66 (1.87) | 5.59 (2.11) | 57.48 (52.0) | 7.11 (2.64) | 21.18 (0.13) | 17.49 (7.06) | 0.65 (0.19) |

These models are used in subsequent analyses and cover 708 heat pumps for the heating curve and coefficient of performance (COP) model, and 637 heat pumps for the utilization model.

entire data set to enhance the interpretability of the subsequent analysis.

**Modeling the heating curve.** The heating curve defines the supply temperature $T_{supp}$ as a linear function of the outdoor temperature $T_{out}$, which most heating controllers allow to be set manually and is known to have a significant impact on performance[17]. Incorporating fixed and random slope and intercept terms, our heating curve model is expressed as:

$$T_{supp}^i(T_{out}) = (a_0^i - 0.270) \cdot T_{out} + (a_1^i + 38.244) \quad (3)$$

When comparing the fixed intercept and slope values to other heating curve models in the literature (e.g.,[59,60]), it becomes evident that these values can be interpreted in the context of a mixed distribution system involving radiators and floor heating. The individual models of each HP either exceed or fall below these baseline values adjusted by the random effects, as they are influenced by their respective distribution system and building insulation level, details of which are unknown in this study.

**Modeling the coefficient of performance.** Theoretically, a COP model could be designed to directly capture deviations from Carnot efficiency. However, this approach is impractical because when there are small differences between the heat source and heat sink temperatures, the denominator of Equation (1) becomes very small. This results in unrealistically high Carnot efficiency values that do not reflect real-world performance. Instead, several studies model the COP as a quadratic or linear function[44,59,61,62]. They either use the outdoor-to-supply temperature difference as a single independent variable or consider outdoor and supply temperatures separately as two independent variables. Note that we use outdoor temperature instead of brine temperature, even for ground-source heat pumps, as they also exhibit a dependence on outdoor temperature. We adopt this approach due to higher data availability, and to eliminate the potential influence of borehole depth on brine temperature measurements. While Fischer et al.[61] and Pospíšil et al.[62] utilize values from multiple HPs reported at operational points in product certificates, Sun et al.[44] employ real measurements but only from a single HP. However, no study has modeled COP using large sample size data sets from multiple HPs in the field. The COP model that performs best on our data set is a simple linear function, given by:

$$COP^i(T_{out}, T_{supp}) = (b_0^i + 0.098) \cdot T_{out} + (b_1^i - 0.104) \cdot T_{supp} + (b_2^i + 6.965) \quad (4)$$

**Modeling utilization as approximation for capacity ratio.** We evaluate the sizing of an HP based on its utilization. To this end, we use the compressor speed of an HP relative to its full-speed capability, expressed as a percentage, as an approximation of an HP's capacity ratio. Since the data set consists of daily aggregates, the average compressor speed for each day is calculated, encompassing total usage, i.e., it includes both space heating and domestic hot water modes. During periods of inactivity, the compressor speed is recorded

as 0%. Similar to the heating curve model, we fit a linear mixed-effects model to describe the utilization of each HP indexed $i$ as a function of the outdoor temperature $T_{out}$, given by:

$$Utilization^i(T_{out}) = (c_0^i - 2.739) \cdot T_{out} + (c_1^i + 50.865) \quad (5)$$

**Evaluating model fits.** To ensure that the interpretation of results is not distorted by potentially poorly fitted models, we only evaluate HPs where the models provide an appropriate fit and where the mixed-effect slopes and intercepts accurately reflect physical properties. Hence, we consider only those HPs where the SMAPE score falls within the interquartile range and HPs with SMAPE $\geq Q_3 + 1.5 \cdot (Q3 - Q1)$ are excluded from the analysis. A root-cause analysis of the reasons for poor model fits of individual HPs is beyond the scope of this paper but could be explored in future research. For example, as outdoor temperatures increase, the supply temperature and utilization of an HP must decrease, while the COP must increase. Due to this condition, 125 HPs (12.21%) are excluded from the heating curve and COP model analysis. Additionally, 190 HPs (18.57%) are excluded due to insufficient data, having fewer than 10 observations of supply temperatures and COP at outdoor temperatures below or equal to 15 °C. As a result, 708 HPs are used for energy efficiency evaluations and for calculating the effects of minor heating curve adjustments. Similarly, for analyses involving utilization models, 174 HPs (17.01%) lacked at least 10 measurements of average compressor speed at outdoor temperatures below or equal to 15 °C, and 212 HPs (20.72%) exhibited an insufficient model fit. Consequently, 637 HPs are included in the sizing evaluations. In contrast, the descriptive analyses in subsequent sections encompass all 1,023 HPs.

Table 1 shows the fits of the corresponding regression models. The values represent the mean and standard deviations of the individual scores of each HP included in the subsequent analyses. Note that, in addition to the individual models, we also provide the score for a combined model, where predictions from a heating curve model replace the original $T_{supp}$ measurements as inputs for a COP model. This approach enables simulations that rely solely on outdoor temperature data. For completeness, the $R^2$ value is also provided, indicating the variance in the data explained by the model. However, note that a small variance in the data can result in a low $R^2$ value without necessarily indicating a poor fit.

**Calculating the seasonal coefficient of performance.** The European standard EN 14825[50] outlines a procedure for calculating the performance of an HP using a single metric known as the seasonal coefficient of performance (SCOP). This standard specifies a set of temperatures and corresponding weights to represent typical temperature conditions across three different climate zones: average, colder, and warmer. For the HPs in Sweden, the Czech Republic, Poland, and Slovenia, we use values corresponding to colder climate conditions. For the single HP in our data set located in France, we use values for warmer climate conditions, and for all other HPs, we assume average climate. The SCOP is determined by taking a weighted average of COP values at these predefined temperatures and is the metric reported on product labels. In addition to outdoor temperatures, the standard specifies

fixed supply temperatures and PLRs. These conditions are rarely met in practical applications without explicit intervention in HP operation, making it impractical to calculate SCOP as strictly defined by the standard[41]. Instead, to accurately assess the real-world performance of HPs, we calculate SCOP under real PLRs and using real supply temperatures obtained from in-situ measurements. We achieve this by sampling from each HP's heating curve and COP model (Equation (3) and Equation (4)) using the fixed outdoor temperatures $T^j_{out}$ and corresponding weights $w^j$ as defined in EN 14825[50] (see Supplementary Table 1). The real SCOP of an HP, indexed by $i$, is thus calculated as follows:

$$\text{SCOP}^i_{real} = \frac{\sum_j \left( w^j \cdot \text{COP}^i(T^j_{out}, T^i_{supp}(T^j_{out})) \right)}{\sum_j w^j} \qquad (6)$$

According to O'Hegarty et al.[41], the value calculated here is comparable to the $\text{SPF}_{H3}$ reported in other studies, but it pertains only to space heating. For completeness, we also report results using the fixed supply temperatures defined in the standard. Although these may not accurately reflect real operating conditions, the calculated SCOP values are closer to those reported on product certificates. In this approach, no sampling from the heating curve is needed; instead, values can be directly sampled from the COP model using the fixed temperatures. Note that in this case, for ground-source heat pumps, the outdoor temperature is fixed at 0 °C, with only the supply temperatures varying, while we continue to use the actual part-load conditions. The complete definition of test points is provided in Supplementary Table 1.

**Simulating minor adjustments to the heating curve.** We simulate a reduction of the heating curve by simply subtracting 1 °C from the intercept. By combining the adjusted heating curve with the original COP model and applying Equation (6), we calculate a new SCOP value. This empowers users and installers to assess the impact on HP efficiency when maintaining the same heat output with lower supply temperatures, providing valuable guidance for optimizing settings. Moreover, this adjustment can be quantified in terms of energy consumption, enhancing its interpretability. Assuming the heat demand $Q_{heat}$ is known, the difference in electricity consumption resulting from a change in the heating curve can be approximated by a function of the old and new SCOP, expressed as:

$$\Delta E = E_{new} - E_{old} = \frac{Q_{heat}}{\text{SCOP}_{new}} - \frac{Q_{heat}}{\text{SCOP}_{old}} \qquad (7)$$

In practice, however, $Q_{heat}$ may not be precisely known due to potential gaps in the measured data. Therefore, we calculate a percentage change relative to the old energy consumption, eliminating the dependency on the exact heat demand as follows:

$$\frac{\Delta E}{E_{old}} \cdot 100\% = \frac{\text{SCOP}_{old} - \text{SCOP}_{new}}{\text{SCOP}_{new}} \cdot 100\% \qquad (8)$$

**Describing the observed performance of all heat pumps**
Figure 1 illustrates the COP values and their temperature dependence across all HPs in our data set, showing only outdoor temperatures at or below 15 °C. This upper limit is consistent with other studies on HP performance, such as[60], and aligns with the European standard EN 14825[50]. We distinguish between air-source heat pumps (ASHPs) and ground-source heat pumps (GSHPs), as well as observations related to the operating modes of SH and DHW production. In the graph, the quantity of observations among several HPs is presented at the top ($N$), while the count of HPs is provided at the bottom ($N_{HP}$). Due to different data availability between operating modes, these figures differ across subsets of samples. The graph reflects two insights that align with

common knowledge in HP literature. The first observation is that GSHPs are generally more efficient than ASHPs - in this case, by approximately 22%, with a mean COP of 4.90 compared to 4.03. This difference is statistically significant at a 99% confidence level, as indicated by the Welch t-test (statistic = -96.28, p-value = 0.0). The reason is that GSHPs do not need to perform defrosting cycles and benefit from stable and higher ground temperatures on cold days, which, although correlated with air temperatures, do not vary as significantly[63,64]. This is also evident in the contour plots shown in Fig. 1c) and d), where a linear interpolation on 500 levels of observed COP over outdoor and supply temperatures is depicted. The efficiency of GSHPs is influenced by outdoor temperature, though not to the extent seen in ASHPs. For GSHPs, the correlation between outdoor temperatures and COP in SH mode is 0.42, whereas for ASHPs, it is 0.49. The second observation is that the COP values are statistically significantly higher for SH compared to DHW, primarily because DHW requires higher flow temperatures[13] (Welch t-test: statistic = 340.81, p-value = 0.0). We address these differences by exclusively comparing HPs of the same type and modeling each HP individually. Furthermore, later assessments of energy efficiency specifically focus on applications related to SH.

**Performance differences among individual heat pumps**
Figure 2 visualizes the performance of HPs, considering the differences between individual systems (also see Supplementary Figs. 1 and 2 for additional graphs). For each specific HP, the median of COP values per operating mode was computed based solely on observations within a defined outdoor temperature range. The charts in Fig. 2 thus display histograms, where each vertical bar represents the distribution of individual HPs within a particular temperature range.

In this context, the $N$-values below the bars indicate the number of HPs used to calculate the proportions. This approach offers a comprehensive overview of the diverse performance and behavior of individual HPs in practice. For instance, while 18.3% of ASHPs still achieve a median COP of 3.0-3.5 at -6 to -3°C, 11.2% fall below 2.0 in this temperature range. Similarly, 11.5% of GSHPs reach a median COP above 5.5 in the temperature range of -3 to 0°C, while an equal percentage fall within the range of 3.0 to 3.5. Thus, HPs can exhibit significant variations in performance, sometimes differing by a factor of 2 to 3 even within the same temperature range, which underscores the importance of identifying low-performing systems.

**Classifying heat pumps in terms of energy efficiency**
Using $\text{SCOP}^i_{real}$ (Equation (6)), HPs can be benchmarked against desired values or compared against each other by applying a distribution-based approach. This allows for the identification of low-performing systems. The Methods chapter provides a detailed explanation and derivation of the thresholds used to classify HP performance, with a brief summary of this procedure below. Existing regulations lack mandatory performance thresholds for HPs in real-world applications, as specified by official policies. However, the EN 14825 standard[50] defines minimum thresholds for the seasonal space heating energy efficiency that HPs should achieve under laboratory conditions during certification. These thresholds can be converted into SCOP limits specific to each HP type, indicating the minimum SCOP below which optimization is required. For HPs exceeding this limit, optimization remains optional but advisable. Additionally, Regulation 811/2013[65] offers a framework to classify HPs from A+++ to G, enhancing interpretability for HP owners to compare categorized performance labels. However, these classifications are again not mandatory for practical applications. The standard distinguishes between HPs designed for low (around 35 °C) and high (around 55 °C) temperature applications, which we average to categorize each HP because the intended application type is unknown in practice. Table 2 categorizes all HPs under assumptions of low, high, or average

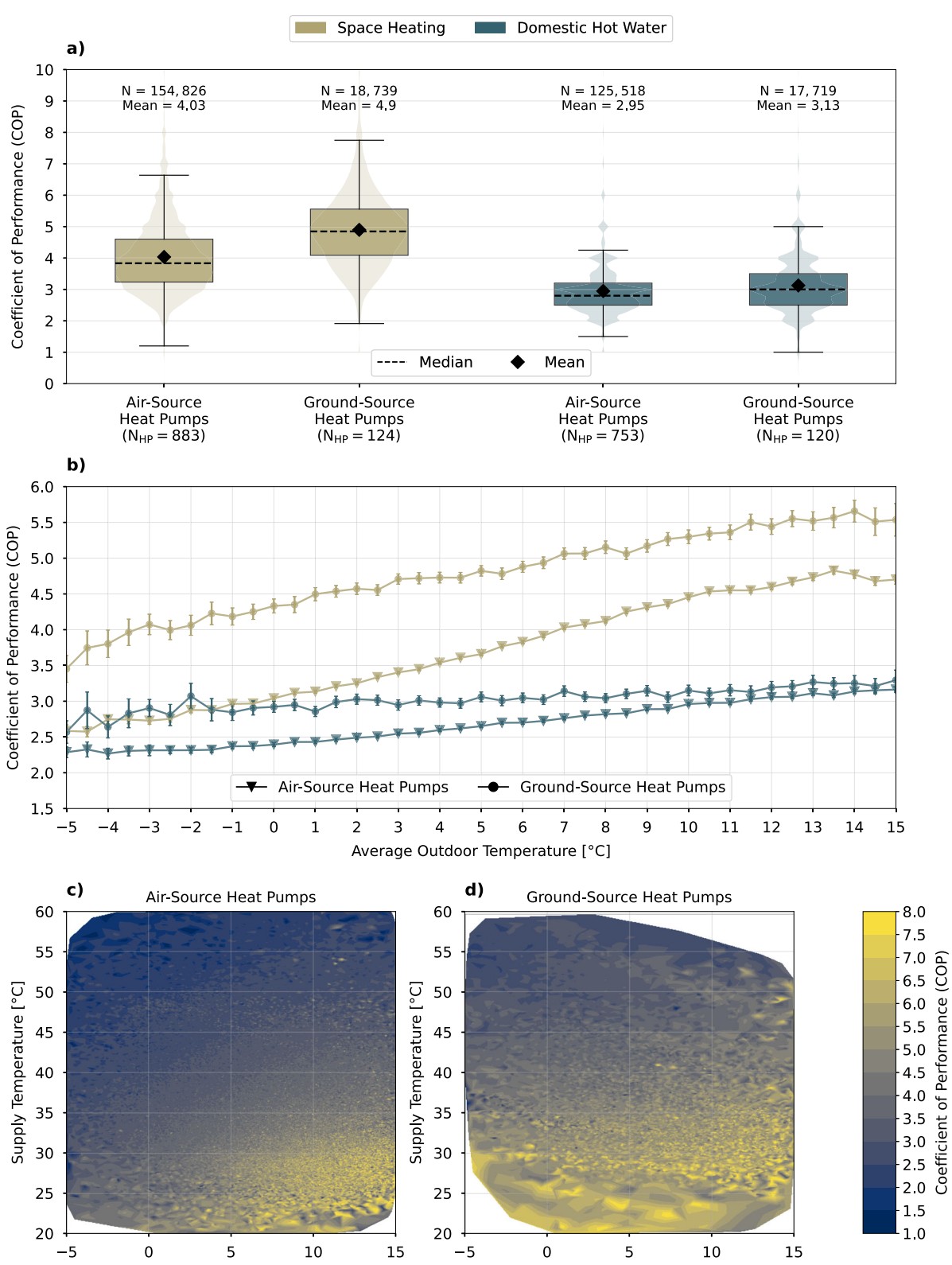

**Fig. 1 | Performance and temperature dependence across heat pump types and operating modes. a** Distributions without accounting for temperatures. The whiskers follow Tukey's original boxplot definition, extending from $Q_1 - 1.5*(Q_3 - Q_1)$ to $Q_3 + 1.5*(Q_3 - Q_1)$, where $Q_1$ and $Q_3$ are the first and third quartiles. **b** Average performance considering outdoor temperature dependence. Error bars represent the 95% confidence interval. **c**, **d** Contour plots with linear interpolations of observed performance across outdoor and supply temperatures. Source data are provided as a Source Data file.

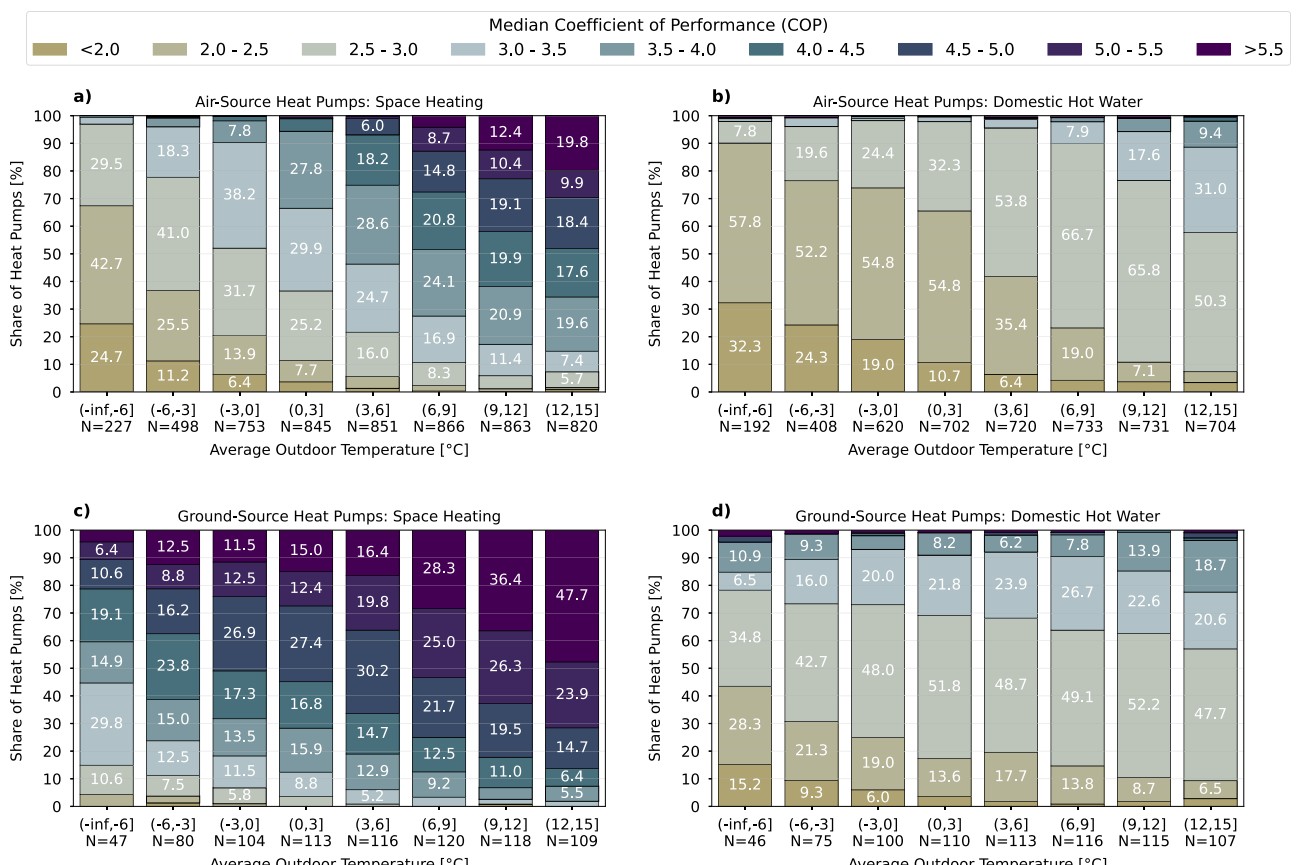

**Fig. 2 | Energy efficiency of individual heat pumps.** A single vertical bar represents a histogram of median coefficient of performance (COP) values calculated for individual heat pumps within specific temperature ranges and different operating modes. Distributions for all: (**a**) air-source heat pumps in space heating mode (**b**) air-source heat pumps in domestic hot water mode (**c**) ground-source heat pumps in space heating mode (**d**) ground-source heat pumps in domestic hot water mode. Source data are provided as a Source Data file.

temperature applications, and we report evaluations with and without fixed supply temperatures. Figure 3 illustrates both the thresholds and results of the classification for average temperature applications, detailed below.

As described in the previous section, 708 systems with sufficient data and appropriate model fits are evaluated, including 612 ASHPs and 96 GSHPs. The average $SCOP_{real}$ of the ASHPs is 3.72, whereas for the GSHPs, it stands at 4.80. The maximum efficiency achieved by an ASHP is 5.55, whereas for GSHP, it is notably higher at 7.36. Among the ASHPs, 17.20% require optimization as they fall below a threshold of 3.01, whereas for GSHPs, only two system (2.10%) fall below the corresponding threshold of 3.14. A significant proportion of HPs achieve high efficiency ratings, with 29.6% (8.3%) of ASHPs (GSHPs) reaching A + level, 30.4% (17.7%) achieving A++ level, and 28.6% (72.9%) even reaching the highest A+++ level. These results underscore that HPs generally exhibit high energy efficiency even in real-world applications. However, the wide range between the lowest and highest performing systems (with a factor of two to three difference in $SCOP_{real}$) highlights a significant performance gap. This underscores the importance of digital monitoring solutions, providing personalized feedback on HP efficiency, and identifying underperforming systems to optimize their operation. Furthermore, it is noteworthy that with 72.9% of GSHPs falling into the highest category A+++, there is a potential need for more refined definitions of classes within the top-performing segment, especially as devices on the market continue to achieve better performance. A further comparison of the observed

performance values with existing field studies utilizing small sample size data sets is provided in Supplementary Note 2.

**Evaluating the effects of adjustments to the heating curve**
Most heating controllers allow the heating curve to be set manually, and as it is known to have a significant impact on performance[17], reducing it is a simple measure to increase energy efficiency with little effort and at low cost (refer to Carnot efficiency in Equation (1)). For this reason, we investigate the effects of lowering the heating curve by shifting it parallel by 1 °C, achieved by a simple subtraction from the intercept (Equation (3)). In Fig. 4, we present the distribution of both the absolute change in SCOP and the relative change in energy consumption (Equation (8)) observed across the 708 HPs.

On average, the SCOP increases by 0.11, and the household energy consumption decreases by 2.61%. This result is consistent with a study with a smaller sample size by Lämmle et al.[66], which analyzed data from 49 HPs and reported that each reduction of one Kelvin increases the seasonal performance factor by 0.10-0.13. With this improvement, 12 ASHPs (11.43% of this category) previously labeled as requiring optimization would now move to the category where optimization is optional, and the same applies to one GSHP (50% of this category). Similarly, 88 ASHPs (14.38% of all ASHPs) and 6 GSHPs (6.25% of all GSHPs) would achieve a better efficiency label. This highlights the substantial impact of HP settings on the energy efficiency achieved in practical applications.

**Table 2 | Statistics and categorization of the actual seasonal coefficient of performance (SCOP$_{real}$) values**

| | | Without Fixed Supply Temperature | | | With Fixed Supply Temperatures | | |
|---|---|---|---|---|---|---|---|
| | | Average Temp. App. | Low Temp. App. | Medium Temp. App. | Average Temp. App. | Low Temp. App. | Medium Temp. App. |
| ASHPs (N = 612) | Mean | 3.72 | 3.72 | 3.72 | 3.97 | 4.48 | 3.47 |
| | Median | 3.72 | 3.72 | 3.72 | 3.98 | 4.45 | 3.49 |
| | Q$_1$ | 3.17 | 3.17 | 3.17 | 3.75 | 4.05 | 3.31 |
| | Q$_3$ | 4.21 | 4.21 | 4.21 | 4.20 | 4.87 | 3.70 |
| | Stdv | 0.71 | 0.71 | 0.71 | 0.38 | 0.58 | 0.41 |
| | Opt. Optional | 507 (82.84%) | 453 (74.02%) | 543 (88.73%) | 605 (98.86%) | 606 (99.02%) | 578 (94.44%) |
| | Opt. Required | 105 (17.16%) | 159 (25.98%) | 69 (11.27%) | 7 (1.14%) | 6 (0.98%) | 34 (5.56%) |
| | A+++ | 175 (28.59%) | 106 (17.32%) | 268 (43.79%) | 188 (30.72%) | 305 (49.84%) | 69 (11.27%) |
| | A++ | 186 (30.39%) | 162 (26.47%) | 185 (30.23%) | 366 (59.8%) | 238 (38.89%) | 446 (72.88%) |
| | A+ | 181 (29.58%) | 197 (32.19%) | 132 (21.57%) | 54 (8.82%) | 64 (10.46%) | 80 (13.07%) |
| | A | 30 (4.9%) | 52 (8.5%) | 18 (2.94%) | 2 (0.33%) | 2 (0.33%) | 7 (1.14%) |
| | B | 23 (3.76%) | 46 (7.52%) | 4 (0.65%) | 1 (0.16%) | 2 (0.33%) | 5 (0.82%) |
| | C | 9 (1.47%) | 16 (2.61%) | 4 (0.65%) | 0 (0.0%) | 1 (0.16%) | 1 (0.16%) |
| | D | 8 (1.31%) | 33 (5.39%) | 1 (0.16%) | 1 (0.16%) | 0 (0.0%) | 3 (0.49%) |
| | E | 0 (0.0%) | 0 (0.0%) | 0 (0.0%) | 0 (0.0%) | 0 (0.0%) | 0 (0.0%) |
| | F | 0 (0.0%) | 0 (0.0%) | 0 (0.0%) | 0 (0.0%) | 0 (0.0%) | 0 (0.0%) |
| | G | 0 (0.0%) | 0 (0.0%) | 0 (0.0%) | 0 (0.0%) | 0 (0.0%) | 1 (0.16%) |
| GSHPs (N = 96) | Mean | 4.80 | 4.80 | 4.80 | 4.80 | 5.52 | 4.29 |
| | Median | 4.84 | 4.84 | 4.84 | 4.92 | 5.44 | 4.38 |
| | Q$_1$ | 4.23 | 4.23 | 4.23 | 4.58 | 5.13 | 3.96 |
| | Q$_3$ | 5.30 | 5.30 | 5.30 | 5.15 | 5.84 | 4.65 |
| | Stdv | 0.87 | 0.87 | 0.87 | 0.57 | 0.74 | 0.54 |
| | Opt. Optional | 94 (97.92%) | 93 (96.88%) | 95 (98.96%) | 96 (100.0%) | 96 (100.0%) | 96 (100.0%) |
| | Opt. Required | 2 (2.08%) | 3 (3.12%) | 1 (1.04%) | 0 (0.0%) | 0 (0.0%) | 0 (0.0%) |
| | A+++ | 70 (72.92%) | 61 (63.54%) | 77 (80.21%) | 85 (88.54%) | 87 (90.62%) | 73 (76.04%) |
| | A++ | 17 (17.71%) | 16 (16.67%) | 17 (17.71%) | 10 (10.42%) | 8 (8.33%) | 17 (17.71%) |
| | A+ | 8 (8.33%) | 17 (17.71%) | 2 (2.08%) | 1 (1.04%) | 1 (1.04%) | 6 (6.25%) |
| | A | 1 (1.04%) | 1 (1.04%) | 0 (0.0%) | 0 (0.0%) | 0 (0.0%) | 0 (0.0%) |
| | B | 0 (0.0%) | 0 (0.0%) | 0 (0.0%) | 0 (0.0%) | 0 (0.0%) | 0 (0.0%) |
| | C | 0 (0.0%) | 1 (1.04%) | 0 (0.0%) | 0 (0.0%) | 0 (0.0%) | 0 (0.0%) |
| | D | 0 (0.0%) | 0 (0.0%) | 0 (0.0%) | 0 (0.0%) | 0 (0.0%) | 0 (0.0%) |
| | E | 0 (0.0%) | 0 (0.0%) | 0 (0.0%) | 0 (0.0%) | 0 (0.0%) | 0 (0.0%) |
| | F | 0 (0.0%) | 0 (0.0%) | 0 (0.0%) | 0 (0.0%) | 0 (0.0%) | 0 (0.0%) |
| | G | 0 (0.0%) | 0 (0.0%) | 0 (0.0%) | 0 (0.0%) | 0 (0.0%) | 0 (0.0%) |

The classification evaluates efficiency classes and determines whether optimization (opt.) is required or optional. It is distinguished between temperature applications (temp. app.), and between evaluations using fixed supply temperatures defined in the standard and original supply temperatures sampled from heating curve models.

## Identifying inappropriately sized heat pumps

The size selection of an HP involves calculating the heating load required for the space it serves and matching it with a system of appropriate capacity[67]. Factors to consider are, for instance, building size, insulation levels, local climate conditions, chosen bivalent temperature, and manufacturer specifications. Properly sizing an HP is critical for maximizing performance throughout its operational life[68]. Despite its importance, installers often lack post-installation feedback on their choices, hindering opportunities for learning and improvement in future installations. Furthermore, detecting undersized systems is crucial to prevent damage. Undersized GSHPs, for example, can extract excessive energy from the ground, potentially leading to permafrost formation around the ground probe and causing it to break[69,70]. Early detection enables adjustments such as integrating additional heat sources to reduce the strain on the system. Accurately evaluating whether an HP is over- or undersized post-installation requires detailed contextual knowledge about building characteristics and design decisions, which is often unavailable in practice. Currently, there is no standardized method to assess inappropriate sizing using field data. However, utilization metrics are valuable indicators in this context, offering insights into HP performance under different conditions[14,57]. High utilization at moderate outdoor temperatures may indicate undersizing, while low utilization in cold conditions suggests potential oversizing. Utilization, expressed as a percentage, allows for standardized comparisons across HP sizes.

By sampling from the utilization model (Equation (5)), we assess the utilization of each HP at critical outdoor temperatures, such as -10 °C (the design temperature for average climate specified in EN 14825[50]) and 16 °C (the heating limit used in EN 14825[50]). These temperatures serve as conservative operational boundaries typically considered for HP performance. For instance, EN 14825[50] specifies -7 °C as the operational limit below which HP manufacturers do not need to guarantee their products' operation, and in well-insulated buildings, the heating limit is generally around 12 °C[15]. Figure 5a) presents a

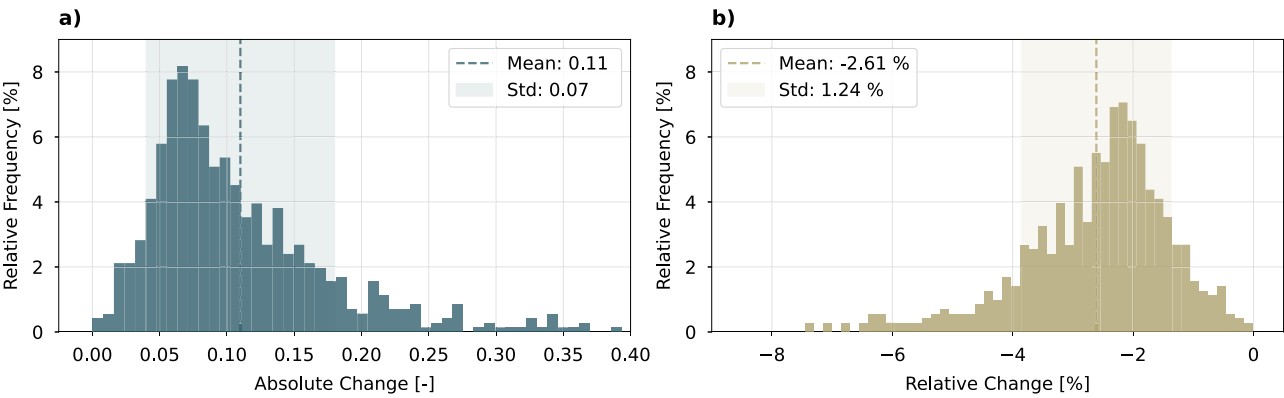

**Fig. 3 | Classification of individual heat pumps by performance.** The heat pumps are categorized using seasonal coefficient of performance (SCOP) thresholds derived from the European standard EN 14825[50] and Regulation 811/2013[65], incorporating a distinction between different heat pump types. **a** Distributions of real SCOP values (see Equation (6)). The whiskers follow Tukey's original boxplot definition, extending from $Q_1 - 1.5*(Q_3 - Q_1)$ to $Q_3 + 1.5*(Q_3 - Q_1)$, where $Q_1$ and $Q_3$ are the first and third quartiles. **b** Evaluation of air-source heat pumps. **c** Evaluation of ground-source heat pumps. Source data are provided as a Source Data file.

**Fig. 4 | Effect of reducing the heating curve by a 1 °C parallel shift on performance.** The graphs illustrate the distributions of the effects for each individual heat pump (N = 708). **a** Absolute change in seasonal coefficient of performance. **b** Relative change in energy consumption. Source data are provided as a Source Data file.

scatter plot illustrating the utilization of each HP sampled at -10 °C and 16 °C, while Fig. 5b) depicts the linear models of each HP alongside the PLR for average climate as defined in EN 14824[50].

Rogeau et al.[57] explored the effects of oversizing through simulation, dimensioning an HP to cover twice the original heating demand at the bivalent temperature. This suggests that an HP operating with 50% utilization at the bivalent temperature may indicate potential oversizing. Applying this criterion to the samples at -10 °C, we find that 43 HPs

(6.75%) show signs of potential oversizing. When assessed at -7 °C, this number increases to 71 HPs (11.15%). Conversely, we identify 5 HPs (0.78%) potentially undersized, as they would still operate with more than 50% utilization at 16 °C. At 12 °C, the assessment identifies 6 HPs (0.94%) potentially undersized. We summarize that inappropriate sizing of HPs may pose a more substantial issue in the field than previously reported in the literature. For example, a study by Weigert[13], which analyzed 228 on-site inspection protocols, reported that only 5% of HPs

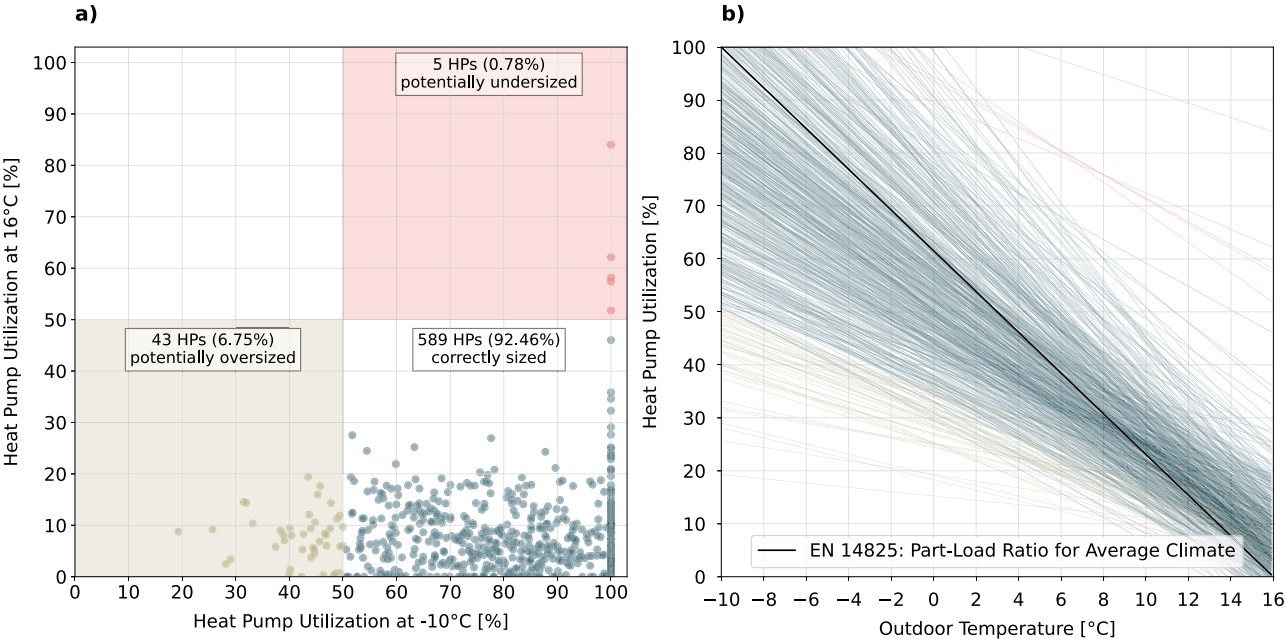

**Fig. 5 | Utilization of individual heat pumps as an indicator for appropriate sizing.** The utilization metric represents the daily average of the compressor speed relative to its full-speed capability, serving as an approximation for the heat pump's (HP) capacity ratio. **a** Evaluation of HP sizing based on utilization, sampled at outdoor temperatures of -10 and 16 °C. **b** Fitted utilization models of each HP ($N = 637$) alongside the part-load ratio for average climate defined in EN 14825 for comparison. Source data are provided as a Source Data file.

were either over- or undersized. In contrast, employing a conservative evaluation approach with our field data reveals approximately 7-11% of oversized HPs and around 1% that are undersized.

## Discussion

With an analysis of 1,023 HPs across 10 countries in Central Europe, this work represents the largest field study on HP performance. Our results and contributions can be summarized in four aspects: Firstly, we deepen the understanding of HP performance in real-world conditions through descriptive analyses and the development of models that enable future studies to simulate HPs based on actual operational data rather than relying solely on product certificates. Our findings reveal significant performance variability among individual HPs, with a 2-3 fold difference between low and high-efficiency systems. Secondly, we operationalize European regulatory thresholds into performance values that can be applied to HP field data, enabling the categorization of HPs into efficiency classes ranging from A+++ to G. Applying these thresholds to the 708 HPs examined, we identified 105 ASHPs (17.2%) and 2 GSHPs (2.1%) operating below the required energy efficiency specifications, underscoring the need for optimization. Further, with 72.9% of GSHPs falling under the highest category (A+++), our work also emphasizes the necessity for improved thresholds derived from real-world operation rather than laboratory conditions, integrated into standardized assessments defined by policymakers. Thirdly, to enhance current operations, we develop a method to evaluate the impact of reducing the heating curve by a 1 °C parallel shift, offering feedback on potential efficiency improvements. Our analysis shows an average improvement in SCOP by 0.11, corresponding to energy savings of 2.61%. With such a minor adjustment in the settings, 12 ASHPs (11.43%) and one GSHP (50%) requiring optimization could meet efficiency thresholds, while 88 ASHPs (14.38%) and 6 GSHP (6.25%) could qualify for improved efficiency labels. This emphasizes the substantial impact of configurations on HP efficiency and underscores the need for digital tools to provide feedback to users and installers. Fourthly, to guide installers in future installations, we propose a method that uses

operational utilization data to assess whether an installed HP is appropriately sized. Even with a conservative evaluation, we find that approximately 7-11% of systems may be oversized and around 1% may be undersized, indicating significant issues in planning and design practices. In the following sections, we discuss the implications of these results for various stakeholders, including policymakers, installers, users, manufacturers, and utilities.

The initial findings on variations in HP performance and the absence of suitable efficiency thresholds highlight the need for enhanced policies to accurately report HP performance, as energy efficiency labels and product certificates are key elements for user guidance[71]. Current certifications, derived from ideal laboratory conditions, often fail to reflect real-world HP operation[15,41]. This issue is similar to the inconsistencies in automotive fuel consumption labeling where lab tests do not capture real-world driving conditions[72,73]. Addressing the misalignment between observed and expected performance is crucial for building public confidence in HP technology and supporting the heating transition, as faster adoption requires positive word-of-mouth[74,75]. Comprehensive post-installation performance standards are urgently needed to bridge the gap in understanding real-world HP performance, especially in diverse building settings[41,50]. A proposed draft to update European regulations aims to make HP monitoring post-installation mandatory but lacks clear criteria for performance evaluation and responsibilities[52]. Closing these gaps is essential to ensure HPs remain economically viable, meet real-world performance expectations, and catalyze broader acceptance among stakeholders, thereby helping to achieve global installation targets[4].

Additionally, the findings on the effects of reduced heating curve configurations are closely linked to the role end-users play in achieving HP efficiency in practice. To this end, more efforts are needed to improve user literacy concerning HP technology, as users with a deeper comprehension of their HPs achieve higher efficiencies[22]. An analysis of the experiences of 83 HP consumers showed that their level of satisfaction depends primarily on operating costs, including both

electricity consumption and maintenance costs[22]. When users were asked about possible improvements, 68% expressed a desire for a control system that provides feedback on cost savings and system efficiency, which underlines their expectation for guidance[22]. Further, troubleshooting through guided user support has been shown to lead to significantly lower maintenance costs compared to engaging energy consultants, technicians or hardware installers, who are also a limited resource[13,76]. This necessity for guidance is also supported by another study[13], reporting that over 40% of users have limited knowledge of the heating control system and require training. The same study identifies that in 57% of the cases, the heating curve setting is set too high and could be reduced[13] and Narayanaswamy et al.[77] report that 40% of modern heating, ventilation and air conditioning systems are generally misconfigured. Addressing this significant prevalence of misconfigurations demands a fundamental shift in approach, necessitating users to possess basic knowledge of maintainability and a willingness to strike a balance between energy efficiency and heating comfort[78]. Instead of opting for excessively high settings to preempt heating comfort issues, installations should incorporate a testing phase. During this phase, settings should be gradually increased from the lowest point until comfort is achieved, balancing it with energy efficiency. Digital monitoring tools that offer feedback on configuration outcomes and demonstrate potential operating cost savings can greatly empower users through education[78].

In addition to enhancing user literacy, it is also imperative to address HP installers, as our results show that many HPs in practice show signs of improper sizing. There is a critical need for enhanced guidance, vocational training, and feedback systems for installers and intermediaries[67], as they constitute both significant drivers and barriers to the transition to energy-efficient and carbon-neutral housing[79–81]. Installers often serve as the primary points of contact for potential HP buyers and as advisors on HP operation. Their influence largely determines whether an HP is installed and, if so, whether it is designed appropriately and which settings are selected. However, installers' perspectives are not neutral, and they tend to opt for what is familiar to them to avoid situations where they lack the necessary skills for installation or advice-giving[82]. Further, related research has shown a poor correlation between installers' estimated and actual energy use of HPs[83]. This is largely due to the complex nature of heat demand calculations incorporating occupant preferences and other factors[83]. To avoid the risk of dissatisfying their customers, many installers tend to overestimate heat demand and choose oversized HPs, which can subsequently reduce operational performance[68,84]. Also, Decuypere et al.[79] report that many installers struggle to keep up with the rapid technological evolution and find it challenging and time-consuming to accurately assess energy efficiency. Digital guided support could offer installers feedback on system design and configurations, helping to optimize the operation of already installed equipment and improve learning for future installations.

To this end, HP manufacturers play a key role in offering such services that enable their appliances to be monitored and controlled[33]. These services should be cost-effective and privacy-preserving in order to achieve broad acceptance. Therefore, the HPs must be designed in such a way that they allow internet-based access to the sensor data. For older HPs where this option is not available in the field, data from the increasingly widespread smart electricity meters offers an alternative with great potential for standardized and manufacturer-independent performance monitoring, as addressed in[14,18,56,85]. This highlights the significant role utilities can play in monitoring HPs, particularly in conjunction with demand response programs and dynamic electricity tariffs. However, also the research community further needs to intensify its efforts on both sensor data and smart meter data to develop methods for HP performance evaluation and feedback. It is crucial that these methods are specifically designed to tackle practical challenges, including the absence of contextual information, handling

inaccurate measurements and data disruptions, and addressing privacy concerns.

## Limitations and future work

This study analyzes data from HPs installed in Central Europe and performance may differ in other geographic regions. Furthermore, we note that the installations are not evenly distributed across the countries included in this study (see details about the real-world data set). To ensure broader generalizability, future studies should validate our results using a data set that includes HPs from various countries and multiple manufacturers, as the current data is derived from a single manufacturer. In addition, the data comes from HPs with internet connectivity. This implies that our methods cannot be used where HPs lack sensors or do not transmit their measurements. In practice, some users may also withhold consent for data analysis due to privacy concerns, particularly regarding the HP's capability to provide real-time occupancy information. As our analyses are focused on SH, future research could extend this work to DHW and cooling applications. Another limitation of this study is that it does not analyze potential programs to exploit dynamic electricity tariffs, which, if prevalent, may influence HP operation. Considering time-of-use would enable field evaluations of the effects of HPs on electricity grids. However, this is beyond the scope of our current study and should be addressed by research focused on flexibility and demand response programs. Furthermore, additional investigations are needed to validate the quantification of inappropriately sized systems in the field, as our study is a starting point to report any figures on this issue. Similarly, we apply efficiency thresholds from European regulations to performance values observed in the field. However, these thresholds were originally intended for use under laboratory conditions. More community efforts are needed to refine these limits to better reflect real-world conditions. Future work could additionally consider integrating contextual details regarding buildings and heating systems, with an emphasis on exploring the utilization of open data for such purposes. This would allow for the use of more sophisticated models and would enable the comparison of HPs in clusters of similar buildings, while maintaining practical relevance. Additionally, real-world applications would benefit from methods for determining individual root causes of inefficient operation to increase user acceptance and provide guidelines for solving the underlying reasons of inefficiency in an automated manner. Nonetheless, this study marks a significant step toward leveraging the potential of digital monitoring solutions for improving energy efficiency of HPs in residential buildings in a scalable manner.

## Methods
### Modeling heat pump performance
For completeness, this section provides an overview of all models tested. Their definitions and parameters are detailed in Table 3. COP Model 6 is inspired by Pospíšil et al.[62] and Fischer et al.[61], modeling COP as a quadratic function of the difference between supply and outdoor temperatures. Similarly, COP Model 1, used by Sun et al.[44], employs a linear model with the same temperature difference. Heating Curve Model 1 follows the definition by Ruhnau et al.[59]. The models selected for further application in this study are Heating Curve Model 1 (see Equation (3)), COP Model 3 (see Equation (4)), and Utilization Model 1 (see Equation (5)). Note that in some models, the dummy variables $d^i_{ASHP}$ and $d^i_{GSHP}$ are used, where $d^i_{ASHP}$ is 1 if the HP indexed $i$ is an ASHP and 0 otherwise, and $d^i_{GSHP}$ is 1 if it is an GSHP and 0 otherwise. These dummy variables allow for modeling even when the HP type is unknown in practical applications. All models incorporate random slopes and random intercepts for each HP, except for COP Model 5, which uses only a random intercept per HP. However, COP Model 5, along with COP Model 2 and COP Model 4, failed to converge, resulting in empty parameter estimates.

**Table 3 | List of all tested and evaluated models**

| Variable | Coefficient | Std. Error | z | P > \|z\| | [0.025, 0.975] |
|---|---|---|---|---|---|
| Heating Curve Model 1: $T^i_{supp}(T_{out}) = (a^i_0 + a_0) \cdot T_{out} + (a^i_1 + a_1)$ | | | | | |
| $a_0$ | −0.270 | 0.010 | −26.239 | 0.000 | [−0.290, −0.250] |
| $a_1$ | 38.244 | 0.241 | 158.915 | 0.000 | [37.772, 38.715] |
| Heating Curve Model 2: $T^i_{supp}(T_{out}) = (a^i_0 + a_0) \cdot T_{out} + (a^i_1 + a_1) \cdot T^2_{out} + (a^i_2 + a_2)$ | | | | | |
| $a_0$ | −0.316 | 0.011 | -28.040 | 0.000 | [−0.338, −0.294] |
| $a_1$ | 0.003 | 0.001 | 5.262 | 0.000 | [0.002, 0.005] |
| $a_2$ | 38.361 | 0.235 | 163.357 | 0.000 | [37.901, 38.821] |
| COP Model 1: $COP^j(T_{out}, T_{supp}) = (b^j_0 + b_0) \cdot (T_{supp} - T_{out}) + (b^j_1 + b_1)$ | | | | | |
| $b_0$ | −0.099 | 0.002 | −57.015 | 0.000 | [−0.103, −0.096] |
| $b_1$ | 6.831 | 0.052 | 131.476 | 0.000 | [6.729, 6.933] |
| COP Model 2: $COP^j(T_{out}, T_{supp}) = (b^j_0 + b_0) \cdot (T_{supp} - T_{out}) + (b^j_1 + b_1) + b_2 \cdot d^j_{ASHP} + b_3 \cdot d^j_{GSHP}$ | | | | | |
| $b_0$ | −0.099 | 0.002 | −43.534 | 0.000 | [−0.104, −0.095] |
| $b_1$ | −19.097 | | | | |
| $b_2$ | 25.734 | | | | |
| $b_3$ | 27.056 | | | | |
| COP Model 3: $COP^j(T_{out}, T_{supp}) = (b^j_0 + b_0) \cdot T_{out} + (b^j_1 + b_1) \cdot T_{supp} + (b^j_2 + b_2)$ | | | | | |
| $b_0$ | 0.098 | 0.002 | 47.833 | 0.000 | [0.094, 0.102] |
| $b_1$ | −0.104 | 0.003 | −37.432 | 0.000 | [−0.109, -0.098] |
| $b_2$ | 6.965 | 0.099 | 70.354 | 0.000 | [6.771, 7.159] |
| COP Model 4: $COP^j(T_{out}, T_{supp}) = (b^j_0 + b_0) \cdot (T_{out}) + (b^j_1 + b_1) \cdot T_{supp} + (b^j_2 + b_2) + b_3 \cdot d^j_{ASHP} + b_4 \cdot d^j_{GSHP}$ | | | | | |
| $b_0$ | 0.098 | 0.002 | 48.771 | 0.000 | [0.094, 0.102] |
| $b_1$ | −0.102 | 0.003 | −40.543 | 0.000 | [−0.107, −0.098] |
| $b_2$ | −426.029 | | | | |
| $b_3$ | 432.762 | | | | |
| $b_4$ | 434.137 | | | | |
| COP Model 5: $COP^j(T_{out}, T_{supp}) = b_0 \cdot T_{out} + b_1 \cdot T_{supp} + (b^j_2 + b_2) + b_3 \cdot d^j_{ASHP} + b_4 \cdot d^j_{GSHP}$ | | | | | |
| $b_0$ | 0.099 | 0.000 | 266.505 | 0.000 | |
| $b_1$ | −0.088 | 0.001 | -156.536 | 0.000 | |
| $b_2$ | −684.681 | | | | |
| $b_3$ | 691.141 | | | | |
| $b_4$ | 692.234 | | | | |
| COP Model 6: $COP^j(T_{out}, T_{supp}) = (b^j_0 + b_0) \cdot (T_{supp} - T_{out}) + (b^j_1 + b_1) \cdot (T_{supp} - T_{out})^2 + (b^j_2 + b_2)$ | | | | | |
| $b_0$ | −0.123 | 0.016 | −7.523 | 0.000 | [−0.155, −0.091] |
| $b_1$ | 0.000 | 0.017 | 0.017 | 0.986 | [−0.032, 0.033] |
| $b_2$ | 7.281 | 0.070 | 103.637 | 0.000 | [7.143, 7.419] |
| Utilization Model 1: $Utilization^i(T_{out}) = (c^i_0 + c_0) \cdot T_{out} + (c^i_1 + c_1)$ | | | | | |
| $c_0$ | −2.739 | 0.041 | −66.214 | 0.000 | [−2.820, −2.658] |
| $c_1$ | 50.865 | 0.620 | 82.025 | 0.000 | [49.650, 52.081] |

The models selected for further use in this study are Heating Curve Model 1 (see Equation (3)), coefficient of performance (COP) Model 3 (see Equation (4)), and Utilization Model 1 (see Equation (5)).

### Deriving a classification scheme for heat pump energy efficiency

To evaluate HPs based on their energy efficiency, we calculate the $SCOP^i_{real}$ (Equation (6)) of each HP according to the definition in the European standard EN 14825[50], which describes performance as a single metric. Note that the definition of SCOP only considers SH, which means that this value is not calculated for DHW. Below, we provide a detailed explanation of how the categorization scheme is derived from European regulations. This scheme distinguishes between HPs where optimization is required or optional, and further categorizes them into distinct efficiency classes. However, it is important to clarify that these regulations form part of the certification and labeling process for HP products under laboratory conditions. Hence, they should not be interpreted as mandatory performance limits that HPs must achieve in practical usage, as such limits currently do not exist.

The standard EN 14825[50] does not specify direct thresholds for SCOP. However, it specifies minimum desired values for the seasonal space heating energy efficiency (SSHEE) $\eta$, expressed as a percentage. According to the definition, SSHEE can be calculated from SCOP as follows:

$$\eta = 0.4 \cdot SCOP \cdot 100\% - F(1) - F(2) \tag{9}$$

The value of 0.4 represents the average European grid power generation efficiency factor. Additionally, $F(1) = 3\%$ serves as a correction factor accounting for contributions from temperature controls, while $F(2) = 5\%$ acts as a correction factor specific to water-to-air or water-to-water systems[41]. As noted in[60], for GSHPs, the combined correction factors $F(1) + F(2) = 8\%$ apply, whereas for ASHPs, only $F(1) = 3\%$ should be used. Consequently, the SCOP can be calculated

based on a given SSHEE $\eta$ as follows:

$$\text{SCOP} = \begin{cases} \frac{\eta + 8\%}{0.4 \cdot 100\%} & \text{for GSHPs} \\ \frac{\eta + 3\%}{0.4 \cdot 100\%} & \text{for ASHPs} \end{cases} \quad (10)$$

According to EN 14825[50], $\eta$ shall not be lower than 110% for typical HP space heaters and HP combination heaters, unless they are low-temperature HPs, for which $\eta$ should not be below 125%. A low-temperature HP is defined as "a heat pump space heater that is specifically designed for low-temperature application, and that cannot deliver heating water with an outlet temperature of 52 ℃ at an inlet dry (wet) bulb temperature of -7 ℃ (-8 ℃) in the reference design conditions for average climate"[50]. Thus, the standard distinguishes between HPs designed for low-temperature applications (supply temperatures around 35℃) and medium-temperature applications (supply temperatures around 55℃)[50]. Using these thresholds as inputs into Equation (10), SCOP values can be derived, below which an HP requires optimization. As a result, GSHPs operating below an SCOP value of 2.95 (medium-temperature) or 3.325 (low-temperature) require optimization. Correspondingly, ASHPs should be optimized if their SCOP falls below 2.825 (medium-temperature) or 3.2 (low-temperature). In a real-world scenario, however, it is often unknown whether an HP was specifically designed for low-temperature applications. Therefore, the choice of the threshold also depends on how rigorous the benchmarking scheme should be. As a compromise, we utilize the average of these SCOP thresholds for each HP type to categorize whether optimization of HPs is necessary or optional. Thus, to evaluate whether an HP requires optimization, we apply a threshold of 3.14 for GSHPs and 3.01 for ASHPs.

Furthermore, the standard EN 14825[50] is complemented by Regulation 811/2013[65], which establishes additional thresholds for SSHEE in the energy labeling of HP space heaters, categorized as A+++, A++, A+, and A to G. Following the same procedure as before, we calculate lower and upper boundaries for each category and HP type. For evaluation, we again utilize the corresponding averages of values from low and medium temperature applications. An overview of all thresholds is presented in Supplementary Table 2, where closed brackets denote that the value is included in the interval, while open brackets indicate that the value is excluded.

## Data availability
The raw data are protected and are not available due to data privacy laws. The processed data and the data generated in this study are provided in the Source Data file. Source data are provided with this paper.

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

## Acknowledgements
This research was funded and supported by the Swiss Federal Office of Energy under the grant number SI/502257 (T.B., T.S., E.F.) and the Bosch IoT Lab at the University of St. Gallen and ETH Zurich (U.P., F.W.).

## Author contributions
T.B.: conceptualization; data curation; formal analysis; investigation; methodology; software; visualization; writing - original draft; writing - review & editing. U.P.: investigation; methodology; software; writing - review & editing. E.F., F.W., & T.S.: writing - review & editing; supervision; project administration.

## Funding

## Competing interests
T.S. & E.F. declare that they are supervisory board members of companies (Hoval and Bosch, respectively) that, among other products and services, manufacture and sell heating systems. T.B., U.P., & F.W. declare no competing interests. The views and conclusions contained in this document are those of the authors and should not be interpreted as necessarily representing the official policy of the sponsors or partners, either expressed or implied. The funding agencies and partners had no control over the design, conduct, data, analysis, review, reporting, or interpretation of the research conducted.
