## [Transparent Peer Review file · Nature Communications]

Estimation of energy efficiency of heat pumps in residential buildings using real operation data

Corresponding Author: Mr Tobias Brudermueller

Version 0:

Reviewer comments:

Reviewer #1

(Remarks to the Author)

Thank you for the opportunity to read and review your very interesting article on “Energy efficiency and behavior of heat pumps in residential buildings under real conditions”. In this article, the authors summarize the results of a large-scale field test in which the efficiency of more than 1.000 heat pumps (HPs) in ten Central European Countries is investigated. Based on the results of this field test, the authors define clusters of HPs according to their efficiency. Moreover, the authors introduce an approach to assess the effects of changes in heating curve settings on efficiency of HPs.

While the data set is impressive and overall, the paper is very well written, there are some major shortfalls that which prevent me from recommending the paper for publication in nature communications:

- (1) The physical/thermodynamical interpretation is lacking. Many of the conclusions drawn by the authors can directly be derived from the Carnot efficiency of a Heat Pump: $COP_{Carnot} = T_{heat_supply} / (T_{heat_supply} - T_{heat_source})$. The main question is: Which factors prevent the HP from working at Carnot efficiency? This would then lead to insights on “bad design” of HPs such as oversizing etc. However, this step is not taken in the analysis.
- (2) In the diagrams results are often mixed: With the broad geographic scope of the data set, there is a large climate impact on the efficiency of HPs. How should an ASHP in an alpine environment have the same efficiency during winter as one that is situated in the Mediterranean area? See Carnot efficiency! Further, ASHPs and GSHPs are often depicted together in one graph, again: The temperature level of the heat sources is per se different. Hence, GSHPs will always have higher SCOPs.
- (3) It is hard to trace the results, as the data set not available, not even in an anonymized/aggregated manner.

Moreover, these points need to be addressed:

- Abstract: The intro in abstract is too long, summary of results falls short
- Abstract: “faulty behavior” By whom? The heat pump or the user? Behavior sounds more like user behavior to me. This is also included in the key words and seems to be misleading, here.
- Abstract: “to identify systems with inadequate sizing or low energy efficiency” What would be main levies identified?
- Keywords: “behavior” and “user feedback” are misleading
- L58: “offer a sustainable alternative” yes, but only if green electricity is used
- L62: “meeting these targets” To my knowledge, there are no binding international targets for HP deployment. There are several national one such as in GE: 6 million by 2030.
- L64f: “poses a significant financial challenge for homeowners”  This argument is one-sided: while coming with high invests, savings during operation occur. The actual question is: when will the system be amortized?
- L65: “These are prevalent”  Vague, I'd rather write “subsidies are”
- L72f: There are two arguments mixed in one: (1) complexity and (2) years of experience and optimization
- L81f: The study by Nolting et al. [14] is rather a simulation study than a field experiment. Hence “in practice” should be rephrased.
- L100: “a country with substantial electricity storage facilities”  how does this contribute to the argument?
- L116f: There are, however, also larger ones: 87 heat pumps by Miara et al. <https://www.ise.fraunhofer.de/en/research-projects/wp-monitor.html>
- L154f: “stems from a single manufacturer”  Impact needs to be discussed in particular with regards to the generalizability of your results.

- L175f: When describing the influence factors on HP efficiency, I strongly recommend to first describe influences according to Carnot efficiency (i.e. temperatures of inlet and outlet) and then factors that lead to deviations thereof.
- L178: This is not a simplification! This is physics ☺
- Fig 1.: Frankly, I deem this figure highly misleading. The drop of COP with higher outdoor temperatures is due to the smaller sample here (i.e. houses with poor insulation that still need heating at these temperatures). This does not say anything about the efficiency per unit.
- L219: Again, this is not "likely due to higher flow temperatures in general". This is a physical fact!
- L225: Did you treat GSHP and water-source HPs the same?
- L234: Again, please have a look at Carnot efficiency.
- L238: And by the mix of heating and cooling in the graph? For "total" you aggregated these groups, right?
- L267f: Putting ASHPs and GSHPs together seems to be a major flaw for me  COPs of GSHPs are to a large extent independent of outdoor temperatures while those of ASHPs are not!  Putting them together in diagrams with outdoor temperature on X-Axis is misleading
- L340: "may show" should rather read "suggests" here
- Fig. 3: When comparing ASHPs for 10 European countries. Should you differentiate according to climate influences? Severe winters in alpine regions vs. mild Mediterranean ones in one graph seem to be misleading.
- L404: "allowing for adjustments in operation"  which ones? Sizing is already done... Rather: Provide insights for futures sizing?
- L433: "Lowering the heating curve by 1°C"  A heating curve consists of gradient and intercept.
- L453ff: One can also directly conduct this from the Carnot efficiency
- L455f: "policy programs may prioritize subsidies for HP installations in new buildings" How can you draw this conclusion? HPs would be less efficient, so more costs to avoid CO₂-emissions. At the same time usually owners with less money  subsidies maybe even more needed than for new buildings, right? Further, do you want to promote technology-specific subsidies, here? You claim to have no conflict of interest.
- L546: Frankly, the study does not offer any insights for HP installers or manufacturers  How to size HP? What is the optimal heating curve for the given building? For low insulated buildings: When are HPs preferable, when conventional heating systems? GSHP vs. ASHP: Which system to use under which circumstances? ETC

Reviewer #2

(Remarks to the Author)

General comments:

The authors highlight that success of the electrification of heat is closely tied to the performance of heat pumps in the field. Given significant electricity demand, demand response via digitalisation will be a component of the heat/electricity intersection. The abstract promises analysis of a large HP sensor data set and a method for comparing the observed performance of individual heat pumps to identify systems with inadequate sizing or low energy efficiency. The author says the method will offer a scalable approach to provide individualized feedback on energy and cost savings and improve performance. The paper addresses a very important research gap. Some analysis of heat pump (HP) data is provided in the form of graphs but without detailed statistical analysis. The method to assess system sizing or efficiency is not sufficiently rigorous to be generalised. The conclusions in Section 3 are not linked to the analysis, but are rather general statements/hypotheses.

The authors highlight the potential total increase in demand in electricity due to HP update, but do not delve further into meaningful analysis. The analysis on HP in situ performance presented is not novel, temperature dependence and poor performance using HP for hot water are well known as well as the better efficiency of GSHPs over ASHPs.

The authors note that HP units are equipped with multiple sensors providing real-time data, but issues of ownership and privacy/ethics nor fairness to the individual versus the aggregate or grid are not mentioned.

The authors provide a nice taxonomy of the real world HP performance papers.

An analysis of the Time of Use/ coincidence of HP demand with other demands and renewable energy source generation would be useful in future research.

It is very unfortunate that the data will not be made available for other researchers to replicate the results and do a deeper analysis.

Some figures are poorly labelled and difficult to read.

Major reservations include:

More information about the HP performance data are needed. Some contextual knowledge is needed about where the sensors are in the system (System boundary), otherwise the analysis and comparisons are not valid.

The data are just over a quarter of a million "daily observations from internal sensor measurements": are these the daily average? Heating season daily averages, or what time step/granularity. A summary of the data description that follows in Section 4 would be better placed before the analysis and results so the reader understands what they are looking at.

HPs can be used for water heating as well as space heating. It's not clear from the analysis if the hot water heating referred to is just legionnaires management, or if all the HP systems are used by the dwellings exclusively for water heating (they may have other hot water supply).

How many ASHP and GSHP are reversible, i.e. able to provide cooling?

What is your definition of "low efficiency"? – check the EU renewable heat definition.

Check the EU Heating degree day definition – the Swiss norm seems much lower.

The article is well written, but the flow is poor with several places where the reader is directed to later sections to find the information (e.g. about the data and the models). Moving section 4 "Methodology" before the results and analysis sections would be a better structure. More detailed statistical analysis is needed, rather than simple graphs and visual analysis.

Minor comments:

P.2 I58 define "natural energy"

P.2 I63 shortfall wrt what target?

P.2 I75 I suggest ">" "...occupant characteristics and HP system settings [12-16]."

P.4 talks about "populations", where "samples" is actually intended.

P.5 Fig 1.c shows the COP versus flow/supply temp delta with ambient outdoor temp – it doesn't look quite right? Please explain.

P.5 the phrase "central heating" is used, do you mean space heating? Otherwise, define the term.

P.6 I267 – Given the difference in performance between ASHPs and GSHPs disaggregating the results by HP type could have been more insightful.

P.7 consider assessing when the COP falls below the EU renewable heat threshold – when/how often does this occur. A deeper analysis is needed. Fig 2.a seems to show 12% of HPs have Cop < 2 when the temperature is in [18-20]. What proportion of HP operation is this associate with – the HP may not be in heating mode at this outdoor temperatures.....I then saw figures e and f in answer. It suggests there's an issue with the flow if the reading is hunting for answers.

P.8 I340 hypothesises that the lack of HP operation suggests the system is oversized. No information about the buildings, occupants, indoor temperature, thermal comfort considerations or electricity prices is provided. There are multiple alternative hypotheses that could be considered.

p.8 I359 Why five groups? Justify your choice and explain how the HPs were clustered. It seems is just ranked by COP quartiles.

P.9/10 Section 2.4 – see earlier comment on system sizing. Backup electric heating is mentioned here. Clarify that in the COP analysis only the electricity for the HP is included in the COP calculation – as noted more information on the HP data and system boundary is needed.

P.10 I434 – see earlier comments about occupants and thermal comfort – while a 1 degree decrease might be beneficial from a performance perspective, no mention is made in the study about engaging with the occupants, not their electricity costs. The analysis about the increased demand takes no account of coincident demand across multiple HPs.

p.11 Fig.4 is poorly labelled.

P.12 I507 The authors conclude that subsidies for buildings with high insulation standards are preferable.....while this is a reasonable statement, there is no information or analysis of the insulation or building energy performance of the dwellings in the study.

P.12 I512 The authors conclude that scalable digital monitoring is essential, but have not show evidence that the digital monitoring in the study is scaleable....nor given details of how the occupants will benefit from the monitoring

p.14 This section contains most of the information the reader needs to know before they try to interpret the results and analysis. Much more detailed explanations is needed about how the data were pre-processed and what type of models were fitted to the data – e.g. simple linear regression models (not curves?). The authors could provide the equations for the models for clarity.

p.16 See the earlier suggestion about using EU renewable heat definitions for your classification.

Reviewer #3

(Remarks to the Author)

This study investigates the energy efficiency and performance of 1023 heat pumps operating throughout Central Europe, bearing considerable significance for the research community. The paper is meticulously crafted. Nonetheless, there are certain issues that need to be addressed.

- Kindly furnish a Nomenclature section encompassing all symbols and abbreviations used.
- In lines 154 to 157, the authors state that all the data are from one manufacturer. While this dataset is valuable, it may yield less representative and potentially biased results. Therefore, it is necessary to address this limitation in the "Limitations and Future Work" section.
- In lines 588 to 591, the calculation of the total heat pumps (HPs) yields 953, which is inconsistent with the total of 1023 HPs. Could you please clarify the whereabouts of the remaining 70 HPs? According to the breakdown provided: 707 air source heat pumps (ASHPs) comprise 69.11%, 186 ground source heat pumps (GSHPs) account for 18.18%, and 60 HPs remain at 5.86%, totaling 953 (93.15%).
- Please verify the accuracy of the percentage calculation: 69.11% versus 69.31%, and 18.18% versus 18.24%. Here is the distribution: Germany: 433 (42.33%), the Netherlands: 210 (20.53%), Austria: 204 (19.94%), Czech Republic: 78 (7.62%), Sweden: (4.50%), Denmark: 35 (3.42%), Poland: 3 (0.29%), Slovenia: (0.20%), France: 1 (0.10%), Great Britain: 1 (0.10%), and unknown: 10 (0.98%).
- The sample distribution among countries seems uneven, with Poland, Slovenia, France, and Great Britain having notably smaller sample sizes, which could make them less representative compared to other countries. It might be prudent to reconsider mentioning all 10 countries in the Abstract to ensure it accurately reflects the sample distribution. Furthermore, this uneven distribution should be acknowledged as a limitation in the "Limitations and Future Work" section.
- Could you please provide the uncertainty of the sensors utilized in this study, as well as an uncertainty analysis for resultant parameters, such as the coefficient of performance (COP)?
- In lines 637 to 639, the second model for COP is unreasonable because COP of the ASHPs is highly impacted by supply air temperature as well as outdoor air temperature. Considering using both supply and outdoor temperatures as inputs for COP regression for ASHPs: $COP = f(T_{supply}, T_{outdoor})$. As for GSHPs, the outdoor air temperature has minor impacts on COP during heating season, therefore it is highly impacted by supply air temperature, the COP for GSHPs can be considered as $COP = f(T_{supply})$.
- To calculate the Seasonal Coefficient of Performance (SCOP) for Air Source Heat Pumps (ASHPs), it's essential to base the calculation on a fixed supply air temperature across various outdoor temperatures. This fixed supply air temperature can be determined by averaging the supply air temperature within the outdoor air temperature range of -10 to 12 °C. For Ground

Source Heat Pumps (GSHPs), the COP is influenced by the supply temperature, which can be inferred from the heating curve at each outdoor temperature. Consequently, the COP at each outdoor air temperature can be calculated accordingly.

Version 1:

Reviewer comments:

Reviewer #1

(Remarks to the Author)

Dear authors,

Thank you for revising the manuscript and for providing detailed answers to each comment. From my point of view, the paper has substantially increased its quality during the revision. In particular by (1) providing the Carnot efficiency and reasoning for derivations thereof, by (2) increasing the statistical analysis of your data set, by (3) providing access to the original data in the supplementary material, and by (4) taking precise account of regional and technical differences when analyzing the extensive data set, the paper has gained a lot in traceability and rigor.

As also stated by the other reviewers in the first round of review, the paper is very well written and addresses an important research topic. Hence, I can recommend the paper to be published in the current version.

Reviewer #2

(Remarks to the Author)

The authors have implemented a very comprehensive revision and addressed the comments of all the reviewers. The revised paper is well written and provides useful insights on the performance of ASHPs and GSHPs. There is a significant gap for "standardized post-installation performance evaluation procedures and digital tools to provide actionable feedback which the authors now contribute to addressing.

Some additional comments:

The paper is well written, one typo in the abstract, (This underscore...), check for others.

The parameters for Eq.s 3 -5 etc are very useful to allow other researcher to perform simulations. I had to route to find them in Table 3. Some minor editing in Section 2 would be useful to point the reader to where they will find these results.

On the method for the model fitting, it is useful to see the comparison of the individual HPs compared to the group, but it is also good practice to split the data into train and test, e.g use 70% of the data to train the models, then evaluate the models on the remaining 30%. This explores the generalizability of the models and reduces overfitting. Since the authors have sufficient data they should evaluate this approach.

p.11 (marked version) check the SCOP formula, the way you've written it doesn't look right, the weights will just cancel out. The wording about the assessment of a change in thermal comfort is still confusing. I think you mean you're trying to assess the impact of a poorer performing HP system to maintain the same heat output? Please clarify.

Reviewer #3

(Remarks to the Author)

The authors have addressed my comments.

Version 2:

Reviewer comments:

Reviewer #2

(Remarks to the Author)

The authors have addressed all my comments and concerns. I recommend the revision be accepted for publication.

Reviewer #1		
R1.1	Overall response: Thank you for the opportunity to read and review your very interesting article on “Energy efficiency and behavior of heat pumps in residential buildings under real conditions”. In this article, the authors summarize the results of a large-scale field test in which the efficiency of more than 1.000 heat pumps (HPs) in ten Central European Countries is investigated. Based on the results of this field test, the authors define clusters of HPs according to their efficiency. Moreover, the authors introduce an approach to assess the effects of changes in heating curve settings on efficiency of HPs. While the data set is impressive and overall, the paper is very well written, there are some major shortfalls that which prevent me from recommending the paper for publication in nature communications:	Thank you for your comprehensive feedback. We sincerely appreciate the time you took to review our paper and your valuable suggestions to enhance our work. Below, we address each comment in detail and outline how we have incorporated your feedback.
R1.2	(1) The physical/thermodynamical interpretation is lacking → Many of the conclusions drawn by the authors can directly be derived from the Carnot efficiency of a Heat Pump: $COP_{Carnot} = T_{heat_supply} / (T_{heat_supply} - T_{heat_source})$. The main question is: Which factors prevent the HP from working at Carnot efficiency? This would then lead to insights on “bad design” of HPs such as oversizing etc. However, this step is not taken in the analysis.	We agree that introducing Carnot efficiency as a key concept is beneficial for readers seeking a deeper understanding of the results. It has now been incorporated into Section 2.2 and is referenced frequently throughout the text to ensure a clear and consistent analysis of its implications.
R1.3	(2) In the diagrams results are often mixed: With the broad geographic scope of the data set, there is a large climate impact on the efficiency of HPs. How should an ASHP in an alpine environment have the same efficiency during winter as one that is situated in the Mediterranean area? See Carnot efficiency! Further, ASHPs and GSHPs are often depicted together in one graph, again: The temperature level of the heat sources is per se different. Hence, GSHPs will always have higher SCOPs.	We now consistently differentiate between air-source and ground-source heat pumps throughout the analysis and in the graphs. Additionally, we address geographical variations by applying tailored analyses based on the specific climate zones in which the heat pumps operate, as defined in the standard EN 14825.
R1.4	(3) It is hard to trace the results, as the data set not available, not even in an anonymized/aggregated manner.	The aggregated data of each individual heat pump is provided as supplementary material. Additionally, the performance scores and parameters for each heat pump’s heating curve model, COP model, and utilization model are included. This allows for future simulations based on real-world measurements rather than product certificates. Finally, the results for each heat pump regarding sizing and efficiency evaluation are also provided.

R1.5	Moreover, these points need to be addressed:	
R1.6	Abstract: The intro in abstract is too long, summary of results falls short Abstract: “faulty behavior” → By whom? The heat pump or the user? Behavior sounds more like user behavior to me. This is also included in the key words and seems to be misleading, here. Abstract: “to identify systems with inadequate sizing or low energy efficiency” → What would be main levies identified? Keywords: “behavior” and “user feedback” are misleading	We have significantly shortened the abstract while incorporating more results. Additionally, we have addressed your comments regarding the terms and keywords.
R1.7	L58: “offer a sustainable alternative” → yes, but only if green electricity is used	Thank you very much for your detailed comments to improve the introduction. We have incorporated all your suggestions and further enhanced the reading flow.
R1.8	L62: “meeting these targets” → To my knowledge, there are no binding international targets for HP deployment → There are several national one such as in GE: 6 million by 2030.	
R1.9	L64f: “poses a significant financial challenge for homeowners” → This argument is one-sided: while coming with high invests, savings during operation occur. The actual question is: when will the system be amortized?	
R1.10	L65: “These are prevalent” → Vague, I’d rather write “subsidies are”	
R1.11	L72f: There are two arguments mixed in one: (1) complexity and (2) years of experience and optimization	
R1.12	L81f: The study by Nolting et al. [14] is rather a simulation study than a field experiment. Hence “in practice” should be rephrased.	
R1.13	L100: “a country with substantial electricity storage facilities” → how does this contribute to the argument?	
R1.14	L116f: There are, however, also larger ones: 87 heat pumps by Miara et al. https://www.ise.fraunhofer.de/en/research-projects/wp-monitor.html	
R1.15	L154f: “stems from a single manufacturer” → Impact needs to be discussed in particular with regards to the generalizability of your results.	Indeed, this needs to be made clearer. Additionally, other reviewers have suggested that the description of the data set should be provided earlier to enhance the reading flow. Therefore, we have relocated the section on the data set to the beginning of the results chapter and emphasized its limitations more strongly.

R1.16	L175f: When describing the influence factors on HP efficiency, I strongly recommend to first describe influences according to Carnot efficiency (i.e. temperatures of inlet and outlet) and then factors that lead to deviations thereof. L178: This is not a simplification! This is physics ;) L219: Again, this is not “likely due to higher flow temperatures in general”. This is a physical fact! L234: Again, please have a look at Carnot efficiency.	As previously mentioned, we agree that this significantly aids in interpreting the results. Thank you for this recommendation. In Section 2.2, Carnot efficiency is explained as a key concept and is now referenced multiple times throughout the text.
R1.17	Fig 1.: Frankly, I deem this figure highly misleading → The drop of COP with higher outdoor temperatures is due to the smaller sample here (i.e. houses with poor insulation that still need heating at these temperatures). This does not say anything about the efficiency per unit.	Indeed, it appears that this aspect was not adequately clarified in the previous manuscript. The samples observed at higher outdoor temperatures correspond to buildings with lower insulation standards that are still requiring heating, whereas others are not, resulting in a reduced sample size. In the revised manuscript, we have now clearly restricted our analyses to temperatures below or equal to 15 degrees Celsius, such that it aligns with the heating limit definition in the standard EN 14825.
R1.18	L225: Did you treat GSHP and water-source HPs the same?	Our data set does not include any water-source heat pumps.
R1.19	L238: And by the mix of heating and cooling in the graph? For “total” you aggregated these groups, right?	In the previous graph, “total” referred to all measurements without distinguishing by operating mode. However, since heating and cooling do not occur simultaneously, “total” actually referred to either cooling and domestic hot water production or heating and domestic hot water production. In the revised manuscript, which now focuses specifically on space heating analyses, we have removed the descriptions of the total and cooling COP. Therefore, this comment is no longer relevant. Future work will include an analysis of domestic hot water production and cooling applications.
R1.20	L267f: Putting ASHPS and GSHPs together seems to be a major flaw for me → COPs of GSHPs are to a large extent independent of outdoor temperatures while those of ASHPs are not! → Putting them together in diagrams with outdoor temperature on X-Axis is misleading	As previously mentioned, we now clearly differentiate between different types of heat pumps throughout the entire analysis. While ground-source heat pumps are still influenced by outdoor temperature, their dependence is less significant compared to the air-source heat pumps (see descriptions in Section 2.3).
R1.21	L340: “may show” should rather read “suggests” here	Thank you, we have revised the wording based on your suggestion.
R1.22	Fig. 3: When comparing ASHPs for 10 European countries → Should you differentiate according to climate influences? Severe winters in alpine regions vs. mild Mediterranean ones in one graph seem to be misleading.	We have updated our approach by differentiating between average, colder, and warmer climate zones per country as defined in EN 14825, rather than using a single average climate zone for all heat pumps. Since the model parameters for each

		system are provided in the supplementary material, future studies can conduct further analyses based on different temperature assumptions.
R1.23	L404: "allowing for adjustments in operation" → which ones? Sizing is already done... Rather: Provide insights for futures sizing?	Thank you for pointing this out; it appears this aspect was not well explained. We now clarify that evaluating appropriate sizing is crucial not only for informing future installations but also for optimizing ongoing operations. In the case of undersized systems, action is required to prevent failures or damage. This can be achieved by integrating additional heat sources.
R1.24	L433: "Lowering the heating curve by 1°C" → A heating curve consists of gradient and intercept.	This point is now clarified: When we refer to "lowering the heating curve," we mean a parallel shift, specifically a subtraction of 1°C from the intercept.
R1.25	L453ff: One can also directly conduct this from the Carnot efficiency L455f: "policy programs may prioritize subsidies for HP installations in new buildings" → How can you draw this conclusion? HPs would be less efficient, so more costs to avoid CO2-emissions. At the same time usually owners with less money → subsidies maybe even more needed than for new buildings, right? → Further, do you want to promote technology-specific subsidies, here? You claim to have no conflict of interest.	We have now generally removed the discussion points regarding the effects of increasing the heating curve and statements related to potential policy programs for retrofitted buildings. We agree that such evaluations require more rigorous assessments that are beyond the scope of this study. Therefore, this comment is no longer relevant for the current manuscript.
R1.26	L546: Frankly, the study does not offer any insights for HP installers or manufacturers → How to size HP? What is the optimal heating curve for the given building? For low insulated buildings: When are HPs preferable, when conventional heating systems? GSHP vs. ASHP: Which system to use under which circumstances? ETC	The questions raised are of high practical relevance, but answering them requires more contextual information about the building, occupants, distribution system, insulation levels, and more, which is currently unavailable to us. We hope to address this in future work. However, our manuscript not only provides important insights into real-world operations using a large sample-size data set but also proposes novel methods for performance evaluation in practical applications. In particular, we are the first to operationalize European regulatory thresholds to enable the categorization of heat pumps into efficiency classes and identify low-performing systems. Furthermore, we have developed a method for identifying heat pumps with either high or low utilization, indicating inappropriate sizing, and providing an indication of how significant this problem is in practice. Additionally, our method allows for simulations of minor adjustments in the heating curve setting, guiding users and installers in finding a good trade-off between energy efficiency and comfort through personalized feedback.

		Further, we hope that by publishing this article in Nature Communications, we can fuel the discussion about real-world heat pump performance and raise awareness of the current issues in heat pump operation. Our work can be a crucial first step toward leveraging digital monitoring solutions for large-scale performance evaluations of already installed heat pumps.
		Thank you once again for helping us improve our manuscript. We are pleased to inform you that we have addressed all your comments and incorporated your feedback. We believe the manuscript has significantly improved, and we hope you share our enthusiasm about the revised version.

Reviewer #2		
R2.1	General comments: The authors highlight that success of the electrification of heat is closely tied to the performance of heat pumps in the field. Given significant electricity demand, demand response via digitalisation will be a component of the heat/electricity intersection. The abstract promises analysis of a large HP sensor data set and a method for comparing the observed performance of individual heat pumps to identify systems with inadequate sizing or low energy efficiency. The author say the method will offer a scalable approach to provide individualized feedback on energy and cost savings and improve performance. The paper addresses a very important research gap.	Thank you for reviewing our work and for the time you have invested. We are pleased to hear that you recognize the importance of our research and agree that it deserves publication. We are pleased to inform you that we have addressed all your comments and incorporated your feedback into the revised manuscript, resulting in substantial improvements.
R2.2	Some analysis of heat pump (HP) data is provided in the form of graphs but without detailed statistical analysis.	We have addressed this issue by adding statistical analyses to aid in the interpretation of the graphs and models, wherever applicable.
R2.3	The method to assess system sizing or efficiency is not sufficiently rigorous to be generalised.	The method to assess appropriate sizing has been completely reworked and subjected to rigorous evaluation. We now clearly explain the link between utilization, the part load ratio, and the heat pump capacity ratio as defined in EN 14825. Based on these definitions, we fit models for each heat pump using compressor utilization as an approximation for the capacity ratio. This approach allows us to sample at both high and low outdoor temperatures to identify systems that may be either over- or undersized. Additionally, the limits for classification are informed by literature rather than being chosen arbitrarily. Detailed descriptions can be found in Sections 2.2 and 2.7.
R2.4	The conclusions in Section 3 are not linked to the analysis, but are rather general statements/hypotheses.	We have revised the summary of our results presented in the first paragraph of the Discussion (Section 3) and integrated them into subsequent statements. As a result, the connection between our findings and the discussion is now more coherent and transparent.
R2.5	The authors highlight the potential total increase in demand in electricity due to HP update, but do not delve further into meaningful analysis.	We consider the argument in the introduction important for understanding why optimizing heat pumps is significant on a larger scale beyond individual cost savings. However, assessing the impact of heat pumps on electricity grids should be included in demand response research and falls outside the scope of this paper. Nevertheless, we have included it in our list of future research endeavors to be addressed (see "Limitations and future work" in Section 3).
R2.6	The analysis on HP in situ performance presented in not novel, temperature dependence	While the aspects you mentioned are not novel, we do not claim them to be. In Section 2.3, we provide

	and poor performance using HP for hot water are well known as well as the better efficiency of GSHPs over ASHPs.	this explanation for completeness and as a foundational basis for less informed users to understand our methods. However, we have significantly condensed this description and emphasized that these are well-known facts in the heat pump literature. The primary contribution of our work lies in providing deeper insights into the real-world performance of heat pumps in the field (Sections 2.3 & 2.4), introducing novel methods for categorizing heat pumps into energy efficiency classes (Section 2.5), techniques for evaluating the impact of modest heating curve adjustments (Section 2.6), and identifying improperly sized heat pumps (Section 2.7). These findings are intended to guide users and installers in enhancing current heat pump operations and to serve as a learning resource for future installations.
R2.7	The authors note that HP units are equipped with multiple sensors providing real-time data, but issues of ownership and privacy/ethics nor fairness to the individual versus the aggregate or grid are not mentioned.	Throughout the paper, we emphasize the importance of addressing privacy concerns to achieve widespread user acceptance of the digital tools we have developed. We further elaborate on these concerns in the limitations section and discuss them as part of future work.
R2.8	The author provide a nice taxonomy of the real world HP performance papers.	Thank you very much for your feedback; we are glad to hear that you found it useful. Additionally, we have included a comparison to existing literature on heat pump performance in the appendix for those seeking a deeper understanding and a more comprehensive overview.
R2.9	An analysis of the Time of Use/ coincidence of HP demand with other demands and renewable energy source generation would be useful in future research.	This is an excellent suggestion. We have included it in the list of future work within the Discussion section (Section 3).
R2.10	It is very unfortunate that the data will not be made available for other researchers to replicate the results and do a deeper analysis.	We provide aggregated and anonymized data for each individual heat pump. Additionally, we offer performance scores and parameters for each heat pump's heating curve model, COP model, and utilization model. This enables future simulations based on real-world measurements rather than relying solely on product certificates, and further ensures the reproducibility of our results. Moreover, detailed results for each heat pump concerning sizing and efficiency evaluation are also included, facilitating deeper analyses.
R2.11	Some figures are poorly labelled and difficult to read.	We have enhanced all graphs in the manuscript and believe this issue is now resolved.
R2.12	Major reservations include:	We are pleased to inform you that we have addressed all your concerns in the revised manuscript.

R2.13	More information about the HP performance data are needed. Some contextual knowledge is needed about where the sensors are in the system (System boundary), otherwise the analysis and comparisons are not valid.	We have now included more detailed information on the data set, specifically addressing the definition of system boundaries and measurement uncertainties, which were previously not covered. The scores reported pertain to the H3 boundary, encompassing the final energy usage of the compressor, fan or brine pump, and the electrical backup heater. Please refer to the updated descriptions of the data set in Section 2.1 for further details.
R2.14	The data are just over a quarter of a million “daily observations from internal sensor measurements”: are these the daily average? Heating season daily averages, or what time step/granularity. A summary of the data description that follows in Section 4 would be better placed before the analysis and results so the reader understands what they are looking at.	We have further clarified this aspect in the data set description, now located in Section 2.1 as per your suggestion, positioned before presenting the results. The observations represent daily averages based on the measurement dates. Although the data set includes observations throughout all seasons, our analyses specifically concentrate on space heating applications, which encompass observations at outdoor temperatures equal to or below 15°C.
R2.15	HPs can be used for water heating as well as space heating. It’s not clear from the analysis if the hot water heating referred to is just legionnaires management, or if all the HP systems are used by the dwellings exclusively for water heating (they may have other hot water supply).	All heat pumps in the data set are utilized for space heating, though not all are also employed for domestic hot water production. Additionally, all heat pumps utilize water rather than air for their distribution systems, which is a common practice in European dwellings. These aspects are now explicitly stated in the updated data set description.
R2.16	How many ASHP and GSHP are reversable, i.e. able to provide cooling?	Out of the 1,023 heat pumps analyzed, 182 units (17.79%) are capable of providing cooling. However, our current analyses are centered on space heating applications, and expanding our methods to include cooling applications is a direction for future research.
R2.17	What is your definition of “low efficiency”? – check the EU renewable heat definition.	Thank you very much for your suggestion; it was very helpful. Instead of using a distribution-based approach to categorize heat pumps in terms of energy efficiency, we now adhere to the thresholds defined in EN 14825 and Regulation 811/2013 to classify heat pumps into classes ranging from A+++ to G.
R2.18	Check the EU Heating degree day definition – the Swiss norm seems much lower.	Thank you once again. We have now incorporated the heating limit of 15°C as defined in EN 14825 into our analysis.
R2.19	The article is well written, but the flow is poor with several places where the reader is directed to later sections to find the information (e.g. about the data and the models). Moving section 4 “Methodology” before the results and analysis sections would be a better structure. More detailed statistical analysis is needed, rather than simple graphs and visual analysis.	Based on your suggestion, we have enhanced the flow of the manuscript to better guide the reader through each section. Although Nature’s guide for authors specifies placing the Methodology section after the Results section, we have reorganized essential parts necessary for understanding the Results to precede the presentation of findings.

R2.20	Minor comments:	We are pleased to confirm that also all minor comments have now been resolved.
R2.21	P.2 l58 define “natural energy”	We now refer to heat pumps that “extract energy from natural sources such as the ground, air, or water.”
R2.22	P.2 l63 shortfall wrt what target?	This is now clarified in the text: the projected shortfall in meeting the International Energy Agency’s global non-binding target of 600 million heat pumps by 2030 is 58%.
R2.23	P.2 l75 I suggest _>“...occupant characteristics and HP system settings [12-16].”	The rephrased sentence in the manuscript is: “The performance of HPs is influenced significantly by factors beyond design, such as occupant characteristics and HP system settings, which is a challenge for manufacturers, installers and owners.”
R2.24	P.4 talks about “populations”, where “samples” is actually intended.	The term “population” has been completely removed.
R2.25	P.5 Fig 1.c shows the COP versus flow/supply temp delta with ambient outdoor temp – it doesn’t look quite right? Please explain.	In heat pump literature, COP is frequently modeled as a function of the difference between supply and outdoor temperature, which is why we selected this type of visualization earlier. However, in the revised manuscript, we now treat outdoor and supply temperatures as two independent variables. Consequently, we have updated the visualizations in Figure 1 accordingly.
R2.26	P.5 the phrase “central heating” is used, do you mean space heating? Otherwise, define the term.	We now consistently use the term "space heating" throughout the entire manuscript, replacing "central heating."
R2.27	P.6 l267 – Given the difference in performance between ASHPs and GSHPs disaggregating the results by HP type could have been more insightful.	Throughout the entire study, we now distinguish between different types of heat pumps and exclusively compare systems of the same type to each other.
R2.28	P.7 consider assessing when the COP falls below the EU renewable heat threshold – when/how often does this occur. A deeper analysis is needed. Fig 2.a seems to show 12% of HPs have Cop < 2 when the temperature is in [18-20]. What proportion of HP operation is this associate with – the HP may not be in heating mode at this outdoor temperatures.....I then saw figures e and f in answer. It suggests there’s an issue with the flow if the reading is hunting for answers.	We reiterate that based on your suggestions, we have significantly improved the flow of the manuscript. Additionally, we have updated the thresholds used for classification based on European regulations.
R2.29	P.8 l340 hypothesises that the lack of HP operation suggests the system is oversized. No information about the buildings, occupants, indoor temperature, thermal comfort considerations or electricity prices is provided. There are multiple alternative hypotheses that could be considered.	We acknowledge that the method for evaluating sizing in the previous version of the manuscript required more rigorous evaluation. We have now completely reworked the proposed methods to align with the definitions in EN 14825 and incorporate insights from additional literature. However, incorporating electricity tariffs into our analyses remains future work, as this information was unavailable to us. Therefore, we have clearly

		acknowledged this limitation in the Discussion section.
R2.30	p.8 l359 Why five groups? Justify your choice and explain how the HPs were clustered. It seems is just ranked by COP quartiles.	This no longer applies to the revised manuscript. Following your suggestion, we now utilize thresholds from European regulations.
R2.31	P.9/10 Section 2.4 – see earlier comment on system sizing. Backup electric heating is mentioned here. Clarify that in the COP analysis only the electricity for the HP is included in the COP calculation – as noted more information on the HP data and system boundary is needed.	Please refer to our response to your comment in R2.13. In this study, the COP pertains to the H3 system boundary.
R2.32	P.10 l434 – see earlier comments about occupants and thermal comfort – while a 1 degree decrease might be beneficial from a performance perspective, no mention is made in the study about engaging with the occupants, not their electricity costs. The analysis about the increased demand takes no account of coincident demand across multiple HPs.	Please excuse the repetition, but incorporating time-of-use and electricity tariffs into our analyses requires additional information that is currently unavailable to us. Nevertheless, we explicitly acknowledge this as a limitation to be addressed in future work.
R2.33	p.11 Fig.4 is poorly labelled.	This issue has been resolved.
R2.34	P.12 l507 The authors conclude that subsidies for buildings with high insulation standards are preferable.....while this is a reasonable statement, there is no information or analysis of the insulation or building energy performance of the dwellings in the study.	We have now generally removed the discussion points regarding the effects of increasing the heating curve and statements related to potential policy programs for retrofitted buildings. We agree that such evaluations require more rigorous assessments that are beyond the scope of this study. Therefore, this comment is no longer relevant for the current manuscript.
R2.35	P.12 l512 The authors conclude that scalable digital monitoring is essential, but have not show evidence that the digital monitoring in the study is scaleable....nor given details of how the occupants will benefit from the monitoring	The next step in our research involves deploying our methods in real-world applications and integrating them into existing service offerings by heat pump manufacturers. This will allow us to gather feedback through interactions with real users. The work presented here marks an initial step toward this goal. The data used relies on simple, low-cost measurements, which are delivered by any modern heat pump with internet connectivity. Furthermore, scalability is guaranteed through the utilization of linear models and the aggregation of data in a daily format, which minimizes the resources required for mass-market applications.
R2.36	p.14 This section contains most of the information the reader needs to know before they try to interpret the results and analysis. Much more detailed explanations is needed about how the data were pre-processed and what type of models were fitted to the data – e.g. simple linear regression models (not curves?). The authors could provide the equations for the models for clarity.	Please refer to the data set description in Section 2.1 and the model descriptions in Section 2.2. We have incorporated all your feedback and have now included formulas for all models and derivations.

R2.37	p.16 See the earlier suggestion about using EU renewable heat definitions for your classification.	As mentioned previously, your feedback on improving the classification scheme was very helpful. We now utilize thresholds based on European regulations.
		Thank you once again for your time and effort in helping us improve our work. We believe that in the revised manuscript, we have addressed all your comments and incorporated your feedback, resulting in significant improvements to the reading flow, methodological rigor, and derivation of highly relevant findings.

Reviewer #3		
R3.1	Overall response This study investigates the energy efficiency and performance of 1023 heat pumps operating throughout Central Europe, bearing considerable significance for the research community. The paper is meticulously crafted. Nonetheless, there are certain issues that need to be addressed.	Thank you for your valuable feedback on how we can improve our paper and for the time you have invested in the review. Your comments have greatly assisted us in the revision process, and we are pleased to report that we have incorporated all of them into the revised manuscript. We hope that you share our enthusiasm for these improvements.
R3.2	Kindly furnish a Nomenclature section encompassing all symbols and abbreviations used.	We agree that having a nomenclature is helpful, so we have added it at the end of the manuscript.
R3.3	In lines 154 to 157, the authors state that all the data are from one manufacturer. While this dataset is valuable, it may yield less representative and potentially biased results. Therefore, it is necessary to address this limitation in the "Limitations and Future Work" section.	The fact that the data comes from a single manufacturer is now included as a limitation in the "Limitations and Future Work" section. For transparency, we have also emphasized this point more strongly in the description of the data set in Section 2.1.
R3.4	In lines 588 to 591, the calculation of the total heat pumps (HPs) yields 953, which is inconsistent with the total of 1023 HPs. Could you please clarify the whereabouts of the remaining 70 HPs? According to the breakdown provided: 707 air source heat pumps (ASHPs) comprise 69.11%, 186 ground source heat pumps (GSHPs) account for 18.18%, and 60 HPs remain at 5.86%, totaling 953 (93.15%).	The confusion regarding the number of heat pumps used for different types of analysis in the previous manuscript has been resolved. Not all heat pumps provide sufficient data to be included in every analysis, requiring exclusion of a subset of samples for certain analyses. For more detailed explanations on this, please refer to Section 2.2, specifically the paragraph titled "Evaluating model fits," which has been newly added.
R3.5	Please verify the accuracy of the percentage calculation: 69.11% versus 69.31%, and 18.18% versus 18.24%. Here is the distribution: Germany: 433 (42.33%), the Netherlands: 210 (20.53%), Austria: 204 (19.94%), Czech Republic: 78 (7.62%), Sweden: (4.50%), Denmark: 35 (3.42%), Poland: 3 (0.29%), Slovenia: (0.20%), France: 1 (0.10%), Great Britain: 1 (0.10%), and unknown: 10 (0.98%).	The rounding errors in the percentages have been corrected, and the numbers have been updated accordingly. Additionally, after conversations with the manufacturer, we have identified the locations of two previously unknown heat pumps. Furthermore, we now have information on the types of all heat pumps.
R3.6	The sample distribution among countries seems uneven, with Poland, Slovenia, France, and Great Britain having notably smaller sample sizes, which could make them less representative compared to other countries. It might be prudent to reconsider mentioning all 10 countries in the Abstract to ensure it accurately reflects the sample distribution. Furthermore, this uneven distribution should be acknowledged as a limitation in the "Limitations and Future Work" section.	Indeed, the distribution of countries is uneven. We have removed the mention of the number of countries from the abstract and have also incorporated this aspect into the limitations section, as per your suggestion.
R3.7	Could you please provide the uncertainty of the sensors utilized in this study, as well as an uncertainty analysis for resultant parameters, such as the coefficient of performance (COP)?	Unfortunately, the specifics of the sensors used are unavailable to us, which hinders detailed calculations of measurement uncertainties. However, the heat pump manufacturer has confirmed that the measurement errors fall within

		the range of the maximum permissible error as per a novel draft for updating European regulations. For transparency regarding uncertainties, please refer to the last paragraph of the data description in Section 2.1, where we have incorporated your feedback.
R3.8	In lines 637 to 639, the second model for COP is unreasonable because COP of the ASHPs is highly impacted by supply air temperature as well as outdoor air temperature. Considering using both supply and outdoor temperatures as inputs for COP regression for ASHPs: $COP = f(T_{supply}, T_{outdoor})$. As for GSHPs, the outdoor air temperature has minor impacts on COP during heating season, therefore it is highly impacted by supply air temperature, the COP for GSHPs can be considered as $COP = f(T_{supply})$.	Thank you very much for this helpful feedback. Although it is common in heat pump literature to model COP as a function of the difference between supply and outdoor temperature, we have considered your comments and now treat outdoor and supply temperatures as two independent variables. This approach not only enhances interpretability but has also significantly improved the models. However, as we demonstrate in Section 2.3, ground-source heat pumps also exhibit correlation with outdoor temperature. Therefore, we include outdoor temperature as an independent variable for ground-source heat pumps as well. In the paragraph "Modeling the coefficient of performance" in Section 2.2, we further explain that using outdoor temperature measurements for ground-source heat pumps is preferable to using brine temperatures due to higher data availability, enabling standardized performance assessments, and eliminating dependence on borehole depth.
R3.9	To calculate the Seasonal Coefficient of Performance (SCOP) for Air Source Heat Pumps (ASHPs), it's essential to base the calculation on a fixed supply air temperature across various outdoor temperatures. This fixed supply air temperature can be determined by averaging the supply air temperature within the outdoor air temperature range of -10 to 12 °C. For Ground Source Heat Pumps (GSHPs), the COP is influenced by the supply temperature, which can be inferred from the heating curve at each outdoor temperature. Consequently, the COP at each outdoor air temperature can be calculated accordingly.	Indeed, the standard for calculating SCOP traditionally defines fixed ambient temperatures and fixed supply temperatures. However, we argue that these fixed supply temperatures often do not accurately represent the real operating conditions of the heat pump. Therefore, in our revised manuscript, we use supply temperatures sampled from the heating curve, as these better reflect real-world operation. Nevertheless, we now include SCOP evaluations for both the real supply temperatures and the fixed test points defined in EN 14825. For more details, please refer to the methodological descriptions in Section 2.2 (Paragraph "Calculating the seasonal coefficient of performance") and the corresponding results in Section 2.5. The full definition of the test points according to the standard is also provided in Table A1 in the appendix to ensure reproducibility.
		Once again, thank you for your feedback on our work and your assistance in improving it. We eagerly await your response to the revised manuscript.

Reviewer #1		
R1.1	Overall response: Dear authors, Thank you for revising the manuscript and for providing detailed answers to each comment. From my point of view, the paper has substantially increased its quality during the revision. In particular by (1) providing the Carnot efficiency and reasoning for derivations thereof, by (2) increasing the statistical analysis of your data set, by (3) providing access to the original data in the supplementary material, and by (4) taking precise account of regional and technical differences when analyzing the extensive data set, the paper has gained a lot in traceability and rigor. As also stated by the other reviewers in the first round of review, the paper is very well written and addresses an important research topic. Hence, I can recommend the paper to be published in the current version.	We are delighted to hear that we have addressed all your comments and that you are recommending our paper for publication. Thank you once again for your valuable feedback, which has significantly strengthened our work.

Reviewer #2		
R2.1	General comments: The authors have implemented a very comprehensive revision and addressed the comments of all the reviewers. The revised paper is well written and provides useful insights on the performance of ASHPs and GSHPs. There is a significant gap for “standardized post-installation performance evaluation procedures and digital tools to provide actionable feedback” which the authors now contribute to addressing. Some additional comments:	We are pleased to hear that you find the paper well written, that it addresses a significant gap in research, and that it provides valuable insights. We also appreciate your additional comments for further improving this work, which have enhanced both the readability of the manuscript and the robustness of our results. As detailed in the following comments, we have incorporated all your feedback into the current version of the manuscript and believe to have addressed all your points. We look forward to your feedback on the revised manuscript.
R2.2	The paper is well written, one typo in the abstract, (This underscore...), check for others.	Thank you for catching this typo. We have thoroughly proofread the entire manuscript once again to ensure there are no additional errors.
R2.3	The parameters for Eq.s 3 -5 etc are very useful to allow other researcher to perform simulations. I had to route to find them in Table 3. Some minor editing in Section 2 would be useful to point the reader to where they will find these results.	Thank you for the suggestion. We have added a sentence in Section 2.2, before introducing the equations, that directly refers to Table 3, guiding readers to the complete statistics.
R2.4	On the method for the model fitting, it is useful to see the comparison of the individual HPs compared to the group, but it is also good practice to split the data into train and test, e.g use 70% of the data to train the models, then evaluate the models on the remaining 30%. This explores the generalizability of the models and reduces overfitting. Since the authors have	We agree that the approach you propose is a valuable addition to the analyses already provided. Accordingly, we have incorporated it into the revised version of the manuscript, specifically in Section A1 of the appendix, with a reference to this type of analysis in Section 2.2.

	sufficient data they should evaluate this approach.	The results indicate only minor differences in performance between models fitted to the entire data set and those fitted to the training data set but evaluated on the test data set. Therefore, we decided to continue using the models fitted on the entire data set, as this allows the performance assessments of each heat pump to be based on all available observations, thereby enhancing interpretability.
R2.5	p.11 (marked version) check the SCOP formula, the way you've written it doesn't look right, the weights will just cancel out.	Thank you for pointing out the lack of clarity. The numerator of the fraction represents a weighted sum of the COP values, which is why the weights do not cancel out with the denominator. To clarify this point, we have added an additional set of brackets to eliminate any confusion.
R2.6	The wording about the assessment of a change in thermal comfort is still confusing. I think you mean you're trying to assess the impact of a poorer performing HP system to maintain the same heat output? Please clarify.	We have revised the description at the end of Section 2.2 and hope this resolves any confusion. The updated text now reads: "This empowers users and installers to assess the impact on HP efficiency when maintaining the same heat output with lower supply temperatures, offering valuable guidance for optimizing settings."
		Thank you once again for the time and effort you invested in reviewing our work. We believe that we have addressed all your comments and incorporated your feedback in the revised manuscript.

Reviewer #3

R3.1	Overall response: The authors have addressed my comments.	Thank you once again for your valuable feedback. We are delighted to hear that we were able to address all your comments.
------	---	---